evolution, ecology, behaviour

bee diversification, floral host shift, floral reward, plant–insect interactions, pollen specialization, zygomorphic flowers

**Author for correspondence:**
Achik Dorchin
e-mail: dorchin@hotmail.com

†Present address: The Steinhardt Museum of Natural History, Tel Aviv University, Tel Aviv 69978, Israel.

# Bee flowers drive macroevolutionary diversification in long-horned bees

Achik Dorchin[1,†], Anat Shafir[2], Frank H. Neumann[3], Dafna Langgut[4], Nicolas J. Vereecken[5] and Itay Mayrose[2]

[1]The National Natural History Collections, The Hebrew University of Jerusalem, Edmond J. Safra Campus, Jerusalem 9190401, Israel
[2]School of Plant Sciences and Food Security, George S. Wise Faculty of Life Sciences, The Edmond J. Safra Center for Bioinformatics, Tel Aviv University, Tel Aviv 69978, Israel
[3]Evolutionary Studies Institute, University of the Witwatersrand, Braamfontein 2050, Johannesburg, South Africa
[4]Department of Archaeology and Ancient Near Eastern Cultures, and The Steinhardt Museum of Natural History, Tel Aviv University, Tel Aviv 69978, Israel
[5]Agroecology Lab, Université libre de Bruxelles (ULB), Brussels, Belgium

AD, 0000-0003-1151-5926; AS, 0000-0001-8059-3544; FHN, 0000-0002-3620-2742; DL, 0000-0002-4824-1044; NJV, 0000-0002-8858-4623; IM, 0000-0002-8460-1502

The role of plant–pollinator interactions in the rapid radiation of the angiosperms have long fascinated evolutionary biologists. Studies have brought evidence for pollinator-driven diversification of various plant lineages, particularly plants with specialized flowers and concealed rewards. By contrast, little is known about how this crucial interaction has shaped macroevolutionary patterns of floral visitors. In particular, there is currently no empirical evidence that floral host association has increased diversification in bees, the most prominent group of floral visitors that essentially rely on angiosperm pollen. In this study, we examine how floral host preference influenced diversification in eucerine bees (Apidae, Eucerini), which exhibit large variations in their floral associations. We combine quantitative pollen analyses with a recently proposed phylogenetic hypothesis, and use a state speciation and extinction probabilistic approach. Using this framework, we provide the first evidence that multiple evolutionary transitions from host plants with accessible pollen to restricted pollen from 'bee-flowers' have significantly increased the diversification of a bee clade. We suggest that exploiting host plants with restricted pollen has allowed the exploitation of a new ecological niche for eucerine bees and contributed both to their colonization of vast regions of the world and their rapid diversification.

## 1. Introduction

The importance of plant–pollinator interactions for speciation has been widely described in the literature, but so far focused on radiation of flowering plants with pollinators mainly being considered as the driving vehicle for their evolutionary success [1–4]. It is generally accepted that pollinator-mediated selection has been central in shaping floral evolution [2]. For example, there is empirical evidence for increased diversification rates in plants with zygomorphic flowers and long corolla tubes or nectar spurs that conceal floral rewards [3–5]. Increased specificity of floral visitors that are associated with specialized flowers has promoted speciation of plant lineages via floral isolation, although other isolating factors also contributed to their speciation [3,6]. Surprisingly, much less is known on how such specialized interactions have shaped the macroevolutionary patterns of floral visitors.

Bees are the group of floral visitors whose evolution and diversification are most closely interrelated with that of the angiosperms [7]. More so than other floral visitors, bees are adapted for the collection and transport of pollen and other floral products, which they require in large quantities for both adult and larval nutrition [8–10]. Flowers with concealed pollen and nectar to control the

removal of rewards by foraging bees are frequently keel-shaped flowers, nototribic flowers and flowers with narrow corolla tubes [9,11–14]. A striking adaptation of bees to these so-called 'bee-flowers' is the evolution of a long tongue (LT), which enabled the efficient extraction of the restricted nectar [15]. Because many bees take nectar from the same flowers that provide them with pollen, LT bees had presumably evolved to exploit both nectar and pollen from bee-flowers. Fossil evidence support this assumption by showing that the appearance of LT bees in the family Apidae coincide with the rapid radiation of more specialized floral configurations, such as bilaterally symmetric zygomorphic flowers [16]. Yet, diverse adaptations to extract restricted pollen are known among modern bees, including also short tongue bees ([17], chapter 19; [18], chapter 7). Distinct behaviours among bees to extract the pollen include direct removal of pollen off the anthers with sometimes modified hairs on the legs, mouth parts, head or thorax, or vibration of flowers to release the pollen, or a combination of these techniques (reviewed by [10,19]). It is anticipated that similarly to plants, increased diversification of bee lineages occurred as a consequence of their association with host plants with restricted pollen. To date, there has been limited empirical evidence to support flower-driven diversification in bees. It has been demonstrated that the transition from carnivory to pollinivory itself had not triggered diversification in bees, and subsequent broadening of host–plant diet was suspected as an important driver [20,21]. Further evolutionary innovations, namely flower buzzing [22] and utilization of floral oils [23], were not significantly associated with increased bee diversification, although they contributed to increase species richness in the former and habitat occupancy in the latter.

One of the largest and widely distributed clades of LT bees are longhorn bees of the '*Eucera* complex' (Apidae) [24]. Species in this clade exhibit large variations in their floral associations, which makes them an interesting model group. While all species may visit bee-flowers for nectar, the females of numerous species may obtain pollen from other types of flowers, and this pollen is the main food resource for their larvae. The early-diverging lineages are associated primarily with actinomorphic, shallow and radially symmetric flowers or inflorescences with accessible pollen, such as those in the Cucurbitaceae, Malvaceae and Asteraceae plant families. Only the more recently diverged subgenera *Synhalonia* and *Eucera* comprise many species that are associated with bee-flowers with restricted pollen such as those frequently found in Fabaceae, Lamiaceae and Boraginaceae (summarized in electronic supplementary material, table S1). This suggests that association with restricted pollen is a derived trait in the *Eucera* complex. Regardless of the pollen type used, oligolectic, pollen specialist species are distinguished from polylectic, pollen generalist species, which obtain pollen from a single or more than one botanical family, respectively [25]. How frequently has the association with restricted pollen evolved, and to what extent had these floral host shifts driven macroevolutionary diversification remains unknown, primarily because of a lack of pollen host spectrum characterization in this group of bees.

This study aims to fill this gap by performing detailed pollen analyses in a phylogenetic context, using a recently proposed phylogenetic hypothesis to trace the evolution of pollen preference in the eucerine clade. Bee preference for dozens of species is partitioned, for the first time, to either 'accessible'

or 'restricted' pollen rewards to test the hypothesis that restricted pollen from bee-flowers has allowed bee lineages to exploit a new ecological niche and diversify at higher rates compared to those that rely on freely accessible pollen.

## 2. Methods

### (a) Taxon sample and dataset

Our dataset comprises the 87 ingroup species from the phylogenetic analyses of the '*Eucera* complex' in [24], and two additional species from the genus *Protohalonia* that were previously not included. This dataset encompasses all species of the early-diversifying genera *Simanthedon* and *Protohalonia*, and representatives of all *Eucera* subgenera, including 19 *Tetralonia* (19.2% of known species), six *Xenoglossodes* (15.4%), six *Peponapis* (42.8%), four *Xenoglossa* (50%), one *Cemolobus* (100%), two *Syntrichalonia* (66.6%), 13 *Synhalonia* (12.4%) and 34 *Eucera* (26.9%). While this selection of 89 species sums up to 22.4% of the 397 *Eucera* complex species (based on [26]), it adequately captures the taxon diversity of this clade, with only four monotypic taxa not represented, which are not expected to alter the phylogenetic tree topology [24]. In all analyses, we used a time-calibrated phylogeny reconstructed with BEAST [27], using nearly the same dataset and the same parameter settings as in [24], the only difference being the inclusion of the two additional species. Accordingly, we used a concatenated sequence alignment of six genes (Opsin + Pol II + NaK + COI + Cytb + 28S) partitioned by codon position and gene region, and four calibration points. The full dataset is presented in the electronic supplementary material, table S1, together with the details of the BEAST analysis.

### (b) Determination of pollen type

We analysed the pollen grains attached to the scopa of female bees to classify each bee species as foraging on either accessible or restricted pollen (see below). We were able to obtain pollen data for 80 of the species included, with only nine species (10%) treated as data deficient. A total of 387 samples and 1–22 replicates per species were obtained from as many different localities as possible in the Nearctic, Palaearctic and Afrotropic regions, which accommodate the vast majority of the eucerine clade diversity (electronic supplementary material, table S1). We followed the methods described by Müller & Kuhlmann [25] to quantitatively analyse the scopal pollen contents. Pollen slides were examined under the light microscope. In heterogeneous samples, percentages of different pollen types were calculated and corrected based on the original size of pollen loads that were assigned prior to the removal of pollen. Pollen types comprising less than 5% of each pollen sample were excluded from the analyses to avoid possible bias from contamination. Pollen grains were identified to the lowest possible systematic level, at least to family, using comparative regional reference collections, in addition to pollen atlases. In plant families that include both accessible and restricted flower morphologies, we further partitioned the corresponding pollen types to either accessible or restricted. When necessary, acetolysis was applied to aid the identification of pollen grains. A detailed description of the quantitative pollen analyses methods is provided as electronic supplementary material and illustration of pollen grains in picture plates S1,S2.

### (c) Ancestral state reconstruction

We used BayesTraits [28] to reconstruct Bayesian ancestral floral preference in the *Eucera* complex. Floral preference was coded for each terminal as either accessible (0), flowers with freely accessible and visible anthers; or restricted (1), flowers with concealed

anthers, not visible to foraging bees. Bee species that carried both accessible and restricted pollen types were coded as (01). The classification of pollen types to restricted and accessible pollen (henceforth abbreviated as 'R' and 'A', respectively) was based on the Plants of the World Online database [29], Zohary [30,31] and Feinbrun-Dothan [32,33] and is presented in electronic supplementary material, tables S1 and S2. In particular, we classified as freely accessible the host plants of the subgenera *Peponapis* and *Xenoglossa*, considered as specialists of Cucurbitaceae and *Cemolobus*, considered as a specialist of *Ipomoea* (Convolvulaceae), based on previous works (cited in electronic supplementary material, table S1). First, we used the Bayesian approach to compute the posterior probability of being at each ancestral state. We then computed Bayes Factor to test if preference for either A or R pollen is significant at each ancestral node.

We performed additional analyses with the terminals partitioned to either pollen specialists (0) or generalists (1). Following Müller & Kuhlmann [25], a species was considered pollen specialist if 95% or more of the pollen grains it carried belong to the same plant family or genus. The details of our ancestral state analyses are presented in the electronic supplementary material.

## (d) Diversification rate analyses

We used the state speciation and extinction (SSE) probabilistic framework [34] to test whether the type of pollen collected by eucerine bees affects diversification rates. Diversification rates for A and R lineages were estimated in maximum-likelihood (ML) analyses, using the SecSSE R package [35], building upon the recent analytical developments of the SSE framework. We applied a series of six models, each with two or four hidden states, which accounts for the possible impact of unobserved traits on the diversification process [36]. All the models were defined with three transition rates: two different transition rates between the observed states and one transition rate between the hidden states. The following models were included in the analysis: (i) $M_a^{se}$, analogous to the full state-dependent BiSSE model, which assumes different speciation and extinction rates only across the observed states (A and R); (ii) $M_a^s$, similar to $M_a^{se}$ but with extinction rates constraint to be equal, thus accounting for the observation that extinction rates are notoriously difficult to estimate and are particularly sensitive to sampling biases [37]; (iii) $M_a^{hse}$, equivalent of the full state-dependent HiSSE model, where the speciation and extinction rates are free to vary among all the combinations of the observed pollen types as well as by binary, uninvestigated, hidden traits [36]; (iv) $M_0$, a null model, in which the extinction and speciation rates of the two pollen types and the two hidden states are constrained to be equal (termed the CR model in SecSSE); (v) $M_0^{h2}$, a more complex null model, comparable to $M_a^{se}$ in terms of parameter richness, in which diversification rates are not assumed to be affected by the pollen types, but possibly by hidden traits (i.e. the speciation and the extinction rates can differ only across the two hidden states); (vi) $M_0^{h4}$, the most complex null model, similar to $M_0^{h2}$, but with four hidden states, and with the same number of free parameters as $M_a^{hse}$. For each model, the likelihood search was optimized using the 'simplex' method with five sets of initial parameters. The best starting point was then used in the final model comparisons. The details of these comparisons are described in the electronic supplementary material.

The ML diversification analyses detailed above were complemented with Bayesian MCMC analyses, implemented in the diversitree R package [38]. These analyses were performed using a five-parameter BiSSE model, in which speciation rates can vary between the two examined states and extinction rates are constrained to be equal, while additionally allowing for unequal transition rates. The results were similar when using a more complex model that additionally allows for state-dependent extinction

rates. An exponential prior distribution was placed on each parameter. The MCMC chains were executed for 10 000 generations with the first 20% discarded as burn-in. We used these analyses to test whether the posterior probability for the net-diversification rate of the R pollen lineages is higher than that for the A pollen lineages ($PP(d_R > d_A)$). The posterior probability that state R exhibits a higher net-diversification rate than state A is represented by the percentage of MCMC samples in which $d_R > d_A$.

In all the analyses, we used a global sampling fraction of 89/397, applying the sampling fraction parameter with equal fractions for both states. To examine the robustness of the diversification analyses to the specific characteristics of our data, we conducted a set of simulations that assessed the power, false-positive rate and accuracy of the estimated diversification rate parameters. The simulations were conducted given a phylogeny and trait data that are similar to those of our empirical dataset. These simulations indicated an adequate false-positive rate for both SecSSE and BiSSE MCMC inference schemes, and that the diversification rate estimates can be accurately inferred. The power to correctly infer differences in diversification rates was high using SecSSE and BiSSE MCMC when extinction rates are constrained to be equal, while the power was lower using the full-BiSSE MCMC inference scheme. We further conducted an additional sensitivity analysis to assess the robustness of the results to different assumptions regarding the trait distribution among the unsampled species ($f$: the fraction of the accessible state among the unsampled species). This analysis indicated that the results are generally robust to different $f$ values, although robustness is more limited in one data partition (PA1, described below), in line with our empirical results (electronic supplementary material, figures S2 and S3, and table S6). Full details of these analyses are provided in the electronic supplementary material.

In all analyses, we compared net-diversification rates in four sets of characters, including three pollen accessibility (PA) sets with alternative categorizations of the polymorphic state: PA1—A versus R pollen rewards, with polymorphic (AR) terminals scored R; PA2—A versus R pollen rewards, with polymorphic terminals treated as ambiguous; and PA3—A versus R pollen rewards, with polymorphic terminals scored A; and a set of pollen specificity (PS)—generalist (polylectic) versus specialist (oligolectic) species, associated with pollen from more than one or a single botanical family, respectively. We included set four to examine the possibility that generalists versus specialists, rather than restricted versus accessible pollen use, drives bee diversification patterns (electronic supplementary material, table S1).

It has been recently shown that diversification analyses could be biased in case the examined phylogeny exhibits trait-related asymmetries in the rate of sequence evolution, leading the state with higher rate of sequence evolution to be incorrectly associated with lower diversification rates [39]. The program TraitRate [40] was thus used to test for such effects. We found that only state A in character set PA3 had a significantly higher sequence evolution rate ($p < 0.05$), and was also associated with a lower diversification rate. However, given that the inferred value of the asymmetry parameter was close to 1.0 ($r = 0.9$), the chance of false-positive inference associating this trait with diversification rates is low, although results should still be interpreted with caution [39].

## 3. Results

## (a) Pollen preference

We have examined the hitherto unknown pollen preference of 80 bee species representing all major *Eucera* complex lineages, which are known to visit various floral host plants. In total, 393 pollen samples assigned to 54 distinct pollen types were

examined in this study, and pollen hosts for 15 additional species were recorded from the literature (figure 1; electronic supplementary material, tables S1 and S2). Asteraceae and Fabaceae were the most abundant pollen hosts, comprising 27.4% and 24% of the pollen recorded, respectively, followed by Cucurbitaceae, 12.5%, and Malvaceae, 5.2%.

Our results show a phylogenetic distribution pattern of pollen hosts, where early-diverging lineages are almost exclusively associated with accessible pollen, and recently diverging lineages associated with restricted pollen, either exclusively or in addition to accessible pollen. Among the 36 early-diverging species sampled (i.e. all lineages except the subgenera *Synhalonia* and *Eucera*), 27 collected almost only (greater than 97% of pollen content) the accessible pollen of Asteraceae, Malvaceae, Onagraceae, Dipsacaceae, Convolvulaceae, Cucurbitaceae and Zygophyllaceae. Nine additional species were polylectic, of which five species collected restricted pollen in addition to accessible pollen (figure 1; electronic supplementary material, table S1).

Among the 44 species sampled from the more recently diverging subgenera *Eucera* and *Synhalonia*, 16 species collected almost only (greater than 86% of pollen content) restricted pollen of either Fabaceae, Lamiaceae, Boraginaceae or Ericaceae. Another species had Fabaceous pollen that could not be determined as being exclusively restricted, and two additional species were polylectic collecting only restricted pollen. By far the most abundant types of restricted pollen were from the Fabaceae (particularly the inverted repeats lacking clade (IRLC); see Discussion), comprising 41.3% of the pollen recorded in these two subgenera. Among the remaining species, 13 were polylectic, collecting significant amounts of accessible pollen, and additional 12 species collected exclusively (greater than 92%) accessible pollen of either Asteraceae, Dipsacaceae or Brassicaceae (figure 1; electronic supplementary material, table S1).

## (b) Ancestral state reconstruction
Ancestral reconstruction of pollen preference indicated that ancient ancestors of the *Eucera* complex were associated with accessible pollen. Ancestors of the early-diverging genera *Simanthedon* and *Protohalonia*, and three other early *Eucera* lineages were inferred with high probability as being associated with accessible pollen (figure 1; electronic supplementary material, table S3). Our analyses suggest that shifts from using accessible to restricted pollen occurred repeatedly and involved four different subgenera (figure 1). Neither preference of accessible nor restricted pollen was recovered as ancestral for *Eucera* and *Synhalonia* (figure 1; electronic supplementary material, table S3). Alternatively, early ancestors in both subgenera probably incorporated restricted pollen into their diets that included both accessible and restricted pollen and preference to either type was then lost in different lineages (figure 1). The shifts to restricted pollen that we recovered as significant were all younger, at the ancestors of Nearctic *Synhalonia* (clade 1 in figure 1) and three species groups within *Eucera*, namely the *longicornis*-, *tristis*- and *vulpes*-groups (clades 2–4, respectively in figure 1; electronic supplementary material, table S3).

The ancestor of the genus *Eucera* and ancestors of all *Eucera* subgenera except *Synhalonia* were inferred as being oligolectic, pollen specialists with high probability (figure 1;

electronic supplementary material, table S3). No polylectic, pollen generalist ancestors were recovered.

## (c) Diversification rate analyses
ML SecSSE analyses showed that among the six SSE models and each of the compared character sets PA1, PA2, PA3 and PS, models with equal extinction rates had the strongest support, with AIC values lower by at least two points from the null (electronic supplementary material, table S4). In all these models, the 'restricted' and 'generalist' states exhibited higher net-diversification rates compared to the respective 'accessible' and 'specialist' states (electronic supplementary material, table S5).

Results obtained using the Bayesian MCMC analyses supported the higher net-diversification rates of 'restricted' compared to 'accessible' pollen-collecting lineages, as reflected by the magnitude of the estimated difference $(d_R{-}d_A)$ and fraction $(d_R/d_A)$ with net-diversification rate greater under the 'restricted' state than under the 'accessible' state, $PP(d_R > d_A)$. The difference in diversification rates was particularly noticeable when using the constrained model with equal extinction rates ($PP(d_R > d_A) > 0.975$ for all PA1, PA2 and PA3 data partitions) and less for dataset PA1 when using the full-BiSSE model ($PP(d_R > d_A) = 0.93$). Results obtained with character set PS indicated that net-diversification rates of generalist lineages were higher although marginally significant than those of specialist lineages, similarly in both models used ($PP(d_G > d_S) = 0.92, 0.93$) (figure 2; electronic supplementary material, table S6).

## 4. Discussion
In this study, we compiled pollen data for 80 eucerine bee species from both the New and the Old World. This allowed us to examine, for the first time, the association of bees to contrasting accessibility of floral rewards. Our results indicate the occurrence of several prominent shifts in floral host use, where derived lineages incorporated restricted pollen from structurally complex bee-flowers into their diet. This is in contrast with ancestral lineages whose diet was based on accessible pollen from mainly actinomorphic flowers. Such floral host shifts were most frequent in the subgenera *Synhalonia* and *Eucera*, but in some of these lineages, the pollen-accessible Asteraceae was maintained as the most common floral host (figure 1). A phylogenetic conservatism of floral hosts is in line with the general trend identified in bees [42]. However, floral host shifts were explained in different groups of bees based on similarity in floral host shapes, quite the opposite from the trend that we highlight [43–45]. Results from additional studies show that shifts between hosts of contrasting floral morphologies are not uncommon and occur even among closely related taxa. This is demonstrated by our results, in bees of the genus *Melitta* (Melittidae [46]) as well as by three studies with megachilid bees that include extensive pollen host datasets [47–49]. Litman *et al.* [20] found that early lineages of Megachilidae that exclusively use actinomorphic, pollen accessible flowers, have lower diversification rates compared to more recent Megachilid lineages. They however considered another behavioural innovation as the main driver of diversification, the inclusion of foreign material in nest construction, the hallmark of the bee family Megachilidae. Our results indicate a clear trend of increased diversification rate in the eucerine clade

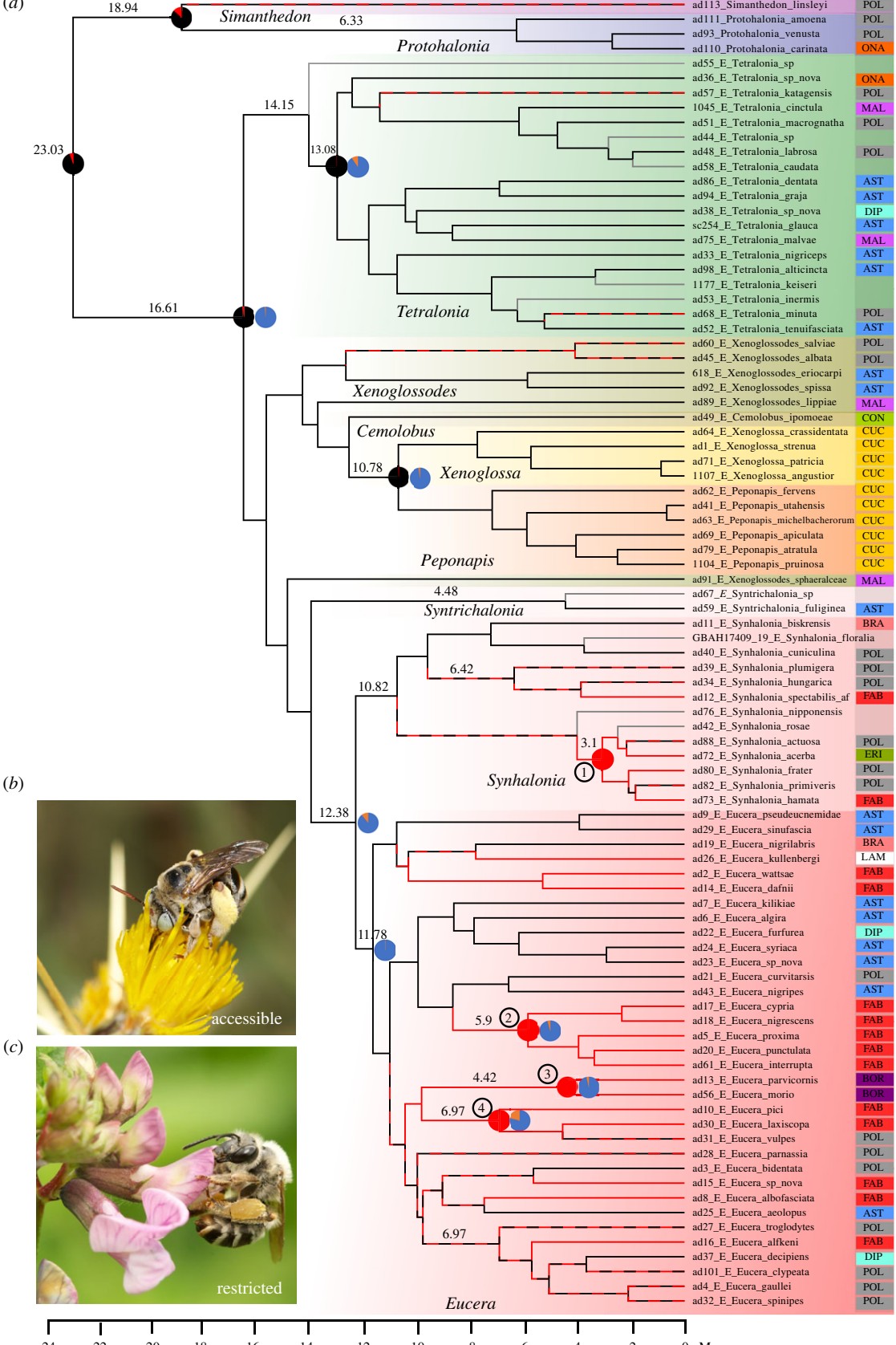

**Figure 1.** The distribution of pollen preferences in the *Eucera* complex. (*a*) Time-calibrated phylogeny with branches coloured according to pollen host accessibility (accessible: black, restricted: red and undetermined: grey), based on parsimony character tracing in MESQUITE [41]. Pie charts at nodes represent probabilities of pollen host accessibility (left charts; accessible: black, restricted: red) and specificity (right charts; specialist: blue, generalist: orange) from ancestral state reconstruction analyses (BF greater than 5). Nodes marked with circled numbers correspond to one *Synhalonia* and three *Eucera* species groups in which a distinct shift to restricted pollen was recovered. Note that only pollen specialist ancestors were recovered. Column to the right of terminals denotes the pollen types partitioned according to the plant host: Asteraceae (AST), Boraginaceae (BOR), Brassicaceae (BRA), Convolvulaceae (CON), Cucurbitaceae (CUC), Dipsacaceae (DIP), Ericaceae (ERI), Fabaceae (FAB), Lamiaceae (LAM), Malvaceae (MAL), Onagraceae (ONA) and Polylectic (POL; pollen generalist). Inferred dates for statistically supported ancestral nodes are given above branches to the left of nodes. Photographs to the left of phylogeny: (*b*) *Eucera* (*Tetralonia*) *graja* foraging on accessible rewards of *Centaurea* (Asteraceae), an oligolectic (pollen specialist) species on Carduoideae; (*c*) *Eucera* (*Eucera*) *nigrescens* foraging on restricted rewards of *Trifolium pratense* (Fabaceae), an oligolectic species on Fabaceae; credit: Nicolas Vereecken. (Online version in colour.)

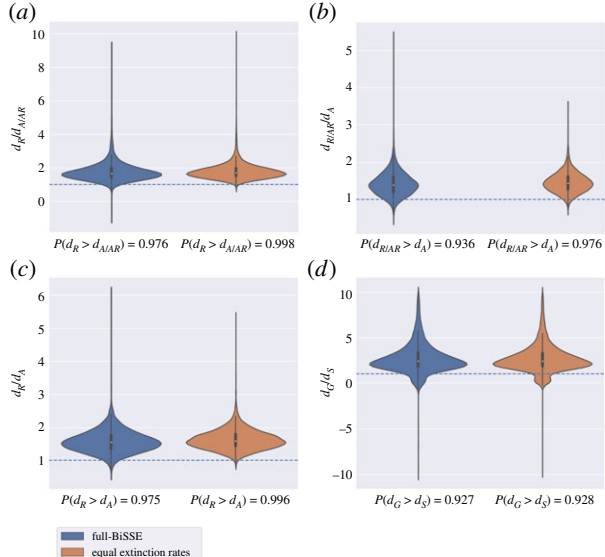

**Figure 2.** Net-diversification rates in the *Eucera* complex. Three sets of traits describing floral hosts with contrasting pollen accessibility (PA1–3) and specificity (PS, described in the methods of diversification rate analyses) are compared. For each set of traits, violin plots (*a*–*d*) represent the distribution of the ratio of net-diversification rates between the compared traits, for (*a*) PA3, (*b*) PA1, (*c*) PA2 and (*d*) PS. (The distribution of the difference between the net-diversification rates are additionally provided in the electronic supplementary material, figure S1). All plots present results from both a full-BiSSE model that allows both the speciation and extinction rates to vary under each character state (left side of each panel) and a BiSSE model with state-dependent speciation and equal extinction rates (right side). The horizontal line indicates the no effect value (i.e. $d_R/d_A = 1$). (Online version in colour.)

following the inclusion of restricted pollen in their diet. This trend was recovered by all analyses, even though the power to detect it was lower when using the full-BiSSE MCMC inference scheme. Using restricted pollen has probably acted to increase bee net-diversification rate by opening an opportunity to exploit a vast food resource as explained below. While our analyses suggest that the difference in net-diversification rates is mainly driven by alterations in the speciation rate and not by extinction rates, these two quantities are hard to tease apart, certainly for the size of the data examined here [37].

The largest group of inaccessible host plants, which had the strongest effect on eucerine diversification belongs to the Fabaceae (electronic supplementary material, table S1). Particularly to the IRLC, which comprises the great majority of the temperate herbaceous Fabaceae, radiating from tropical and subtropical regions into the temperate zone [50,51]. Among the largest IRLC genera that diversified during the Miocene are *Vicia* (approx. 160 species) and *Lathyrus* (approx. 160), *Trifolium* (approx. 250) and *Astragalus* (approx. 3000), dated *ca* 17.7, 16–23 and 12.5 Ma, respectively [51–53]. Dorchin *et al.* [24] inferred the more recent diversification of the subgenera *Eucera* and *Synhalonia* at 12.3–13.9 Ma (see also figure 1), an age which parallels the radiation of these IRLC taxa and coincides with their historical biogeography. *Eucera* and *Synhalonia* diversified in temperate regions of the Holarctic [24], where the IRLC taxa are abundant and widespread, and we show they developed an association with many of these plants as preferred or exclusive host plants. By contrast, the earlier diversification of the subgenera

*Tetralonia* and *Xenoglossodes*, and their occurrence in warm regions of the Old and the New World, respectively [24], suggest a smaller overlap with that of the IRLC, and accordingly, they are associated mainly with Asteraceae and other pollen accessible flowers.

Our results provide contrasting evidence to the hypothesis that broadening of pollen diet, in addition to floral accessibility, increased eucerine diversification rate. Such an effect supports the hypothesis that broadening of pollen host diet was the main factor that triggered diversification in more recent bee lineages relative to primitive bee lineages, which are mostly pollen specialists (oligolectic) [21]. Our results demonstrate the difficulty to extrapolate this general trend to particular taxa due to the labile nature of pollen host range in bees ([18], chapter 12). Shifts from oligolecty to polylecty are distributed across our phylogeny and may represent reversals, because the first branching, species-poor genera *Simanthedon* and *Protohalonia* are mostly polylectic (figure 1). Notably, a more thorough sampling than the one we have done is required to accurately determine the pollen spectrum of a species. Furthermore, floral preference in bees reflects a continuum of dietary specialization rather than dichotomy [25,54] as with PA, and this may result in oversimplification of pollen specialization effects.

Another factor that could have possibly influenced eucerine diversification rate and was not examined in our study is pollen chemistry [55–58]. Pollen chemistry is relevant to our study because some common types of accessible pollen in our dataset are notoriously known as a poor source of nutrition to bees. Thus, our results could be explained by shifts from nutrient-poor pollen of Asteraceae and Malvaceae to Fabaceae, and other types of pollen like Cucurbitaceae that typically have higher protein : lipid ratios [59]. Asteraceae pollen has repeatedly hampered the development of both solitary and social bee preimaginal stages, and this was attributed to low protein contents, the lack of essential amino acids, difficult digestibility and the presence of unfavourable sterols and secondary pollen compounds [25,58,60–63]. Müller & Kuhlmann [25] further suggested that the freely available Asteraceae pollen is protected chemically by secondary metabolites such as pyrrolizidine alkaloids to prevent from overexploitation by generalist bees that are not adapted to digest the pollen. This assumption is supported by the high incidence of Asteraceae bee specialists and the scarcity of closely related generalists that use Asteraceae as host plant [25,54]. Interestingly, we found that eight polylectic *Eucera* species from four different subgenera collected significant amounts of Asteraceae pollen (8% averaged content or more; electronic supplementary material, table S1). These polylectic species are derived from a clade with many Asteraceae specialists and may have been preadapted to digest Asteraceae pollen [42,48]. It is possible that chemical protection in pollen accessible hosts was replaced to some extent by physical protection in restricted pollen hosts of eucerine bees. This could have released the bees from the physiological constraints required for coping with the unfavourable properties of the pollen and contributed to speciation. However, some types of restricted pollen, including in Fabaceae and Boraginaceae, were found to be chemically protected [49,56,64,65]. Our diversification analyses suggest that such unobserved hidden traits in our system had a smaller effect than that of PA on eucerine diversification rate.

# 5. Conclusion

Our study provides the first empirical evidence to show that floral host shift has driven increased diversification of a bee clade. The ability to extract pollen from bee-flowers has opened a new ecological niche for eucerine bees and contributed to their distribution and diversification in vast regions of the World. The evolutionary pattern described here is a strong incentive for additional studies on flower-bee associations and their effect on the evolution, diversification, and behavioural and morphological adaptations of bees.

Data accessibility. The R scripts used to conduct all diversification analyses and additional supporting data are available from the Dryad Digital Repository: https://doi.org/10.5061/dryad.h18931zmg [66].

The data are provided in the electronic supplementary material [67].

Authors' Contributions. A.D.: conceptualization, data curation, formal analysis, funding acquisition, investigation, methodology, project administration, resources, software, supervision, validation, visualization, writing-original draft, writing-review and editing; A.S.: data curation, formal analysis, funding acquisition, investigation, methodology, resources, software, validation, visualization, writing-original draft, writing-review and editing; F.H.N.: data curation, investigation, methodology, resources, validation and visualization; D.L.: data curation, investigation, methodology, resources, validation and visualization; N.J.V.: formal analysis, investigation, methodology, software, validation and visualization; I.M.: data curation, formal analysis, investigation, methodology, resources, software, supervision, validation, visualization, writing-original draft, writing-review and editing.

All authors gave final approval for publication and agreed to be held accountable for the work performed therein.

Competing interests. We declare we have no competing interests.

Funding. This work was supported by a Vatat Training and support fellowship at the Hebrew University of Jerusalem to A.D., and a fellowship from Edmond J. Safra Center for Bioinformatics at Tel Aviv University to A.S.

Acknowledgements. We thank Valentina Epshtein, Carlos Cordova, Louis Scott, Jennifer Larson and Andreas Müller for help with pollen identification; Stella Watts and Thomas Wood for unpublished pollen analyses; Mark Pagel for assistance with BayesTraits; Terry Griswold, Harold Ikerd, Connal Eardley, Sarah Gess, Alain Pauly, Martin Schwarz and Esther Ockermüller for providing bee specimens. Prosper Bande has processed bee pollen loads at Evolutionary Studies Institute and mounted the slides for analysis. We are grateful to three anonymous reviewers and the journal subject editor for many comments that considerably helped to improve this paper.

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
