## [Peer Review File · Proceedings of the Royal Society B: Biological Sciences]

Review History

RSPB-2020-2003.R0 (Original submission)

Review form: Reviewer 1

Recommendation

Major revision is needed (please make suggestions in comments)

Scientific importance: Is the manuscript an original and important contribution to its field?

Excellent

General interest: Is the paper of sufficient general interest?

Excellent

Quality of the paper: Is the overall quality of the paper suitable?

Acceptable

Is the length of the paper justified?

Yes

Should the paper be seen by a specialist statistical reviewer?

No

Do you have any concerns about statistical analyses in this paper? If so, please specify them explicitly in your report.

No

It is a condition of publication that authors make their supporting data, code and materials available - either as supplementary material or hosted in an external repository. Please rate, if applicable, the supporting data on the following criteria.

Is it accessible?

Yes

Is it clear?

Yes

Is it adequate?

Yes

Do you have any ethical concerns with this paper?

No

Comments to the Author

Review of Dorchin et al.

This study is great effort to get at a question which ought to be interesting to a great deal of researchers. As a person strongly on the plant-side of pollination, I was unaware that there was little work on diversification of bees in response to floral morphology. The pollen data which they assembled was surely a great deal of work and the choice of a group of bees and the sampling (20% of the clade) is quite high for these larger-scale ecological comparative analysis. I think this paper is certainly of the suitable for publication in Proc B after revisions and I am sure it will be of interest to many. I have pages of individual comments, but they broadly fall into:

- 1) Clarity issues – there are a lot of methods and results which are not explained sufficiently to understand exactly why you chose to do them. For instance, why is pollen volume used and how is it calculated from the percentages gathered? And if the bee was collected early in a foraging bout, why should a lower volume of pollen be any less indicative of preference?
- 2) There are a few long paragraphs with multiple ideas that need to be shortened and clarified (i.e. 376-417).
- 3) Many references are needed throughout for various statements and in some cases, references should be checked for accuracy (i.e. final paragraph).
- 4) Certain word choice is vague and needs to be made much clearer: e.g. “Evolutionary success” is used to indicate an increase in diversification rates.
- 5) The final paragraph about the evolution of the specialist group and the flowers they rely on needs to be significantly rethought and re-referenced.

Title: It seems like the novelty of this study is that you focus on the bees, yet the title says “bee flowers” first, and until I read the abstract, I thought this would be a plant paper. I’d rephrase somehow to stress the bee aspect. (Even, perhaps, just reversing bees and bee flowers).
 30-31: I’d be a little more circumspect here (and elsewhere) with word choice – you don’t really know that these bees (or any floral visitor!) is truly a pollinator, or if a pollinator, a good one or one important to the plant. You are very careful with this elsewhere (i.e. the nice system/study description in L95-115). It is worth being very specific, even in these broad sections like L30-34, as

the bees needn't be pollinators to have diversifying in response to floral preference/floral traits. "Floral visitor" is the phrase I use most often, but that's not really appropriate here – perhaps "pollen consumer" or something similar. Loads of even specialist bees/moths/etc, are not particularly good pollinators, but are reliant on one or few plants.

55-56: The clause "and many floral traits are used in the characterization of angiosperm species" is a non sequitur here, and interrupts the flow between pollination ideas. I'd delete or rephrase this sentence.

57: What exactly is the assumption here? That pollinators shape floral evolution? I think deleting the clause will make this clearer, but it might be worth rephrasing this section.

56-65: You definitely don't have to take this suggestion, but in the two sentences before the concluding one, you discuss a lot of pollinator specialization (which implies some evolutionary knowledge), then say that there is little known about the evolution of pollinators. As I understand the literature, the diversification of plants after developing things like nectar spurs or long tubes is not actually demonstrated from pollinator specialization (i.e. most hummingbirds and hawkmoths are not specialists, in the sense that they hit a single species or genus of flowers), instead from these plant morphological changes decreasing the generality of their flowers; i.e. the plant is specializing, not really the pollinator (which you get at in L58-59), but then you switch to pollinator specialization and repro isolation in the next, and conclude that little is known about evolution of pollinators.

76-78: Might be nice to have a reference here: bees could certainly exploit flowers for both pollen and nectar without being long-tongued or co-evolving with the flowers (lots of native bees here nectar and collect pollen of non-native flowers, including restricted zygomorphic species, with which they have no co-evolutionary history at all).

81-83: This sentence about exceptions in bees, needs context, since it follows a plant sentence. What exact exceptions do you mean? What flowers they exploit? The bee morphology (i.e. non-LT bees exploiting zygomorphic flowers)?

94: "potential evolutionary advantage" What do you mean by this statement? Later on, in your results, you use this to mean increased diversification, but this is distinctly not what you mean here. Rephrase or define exactly.

95-115: This is a great overview, but can you detail a little bit of their interaction (i.e. they collect pollen, do they also drink nectar? You collect pollen from the scopa later, but you mention nototriby in the intro – do they also get pollen in other parts of the body or do they groom it down right away?). This is probably all known by most bee people, but this paper should attract attention of a lot of plant people, too, so a little more natural history of the bees would be nice for those of us who don't know bees as well.

102: Boraginaceae are actinomorphic and often shallow. I know that some are deep and specialized, but many readers will probably picture something like *Heliotropus* or *Nama*, which are actinomorphic and probably easily exploited by many floral visitors.

119: How many species total are there in *Eucera*? For the 85 species, how much of the radiation do they cover? This is at first glance a good sample size for comparative analyses (especially that you sampled pollen from 76 of the 85 species!!!), but it is worth taking the cautions detailed in Losos (*American Naturalist*, 2011, 177: 709-727) into consideration: i.e. if those 85 are drawn from across a phylogeny of 1500 species, there are some caveats to that analysis (in L95-96, you just write "one of the largest", and as a plant person, I have no idea if this is 100, 500, or 1000 and that surely matters to interpretation). Later on you clarify that it is 20%, but this needs to be noted here, too.

131-134: What a dataset!

139: Why are percentages per species summed? I do not think you can sum percentages (i.e. 50% + 75% is uninterpretable); did you mean that they were they averaged? If that is what you mean, is this appropriate without some sort of weighting (i.e. if you had 1000 pollen grains on one bee, should that be averaged with another bee of the species which had 10 pollen grains?)

138-145 (and again 262-279): It seems as if you are calculating percentages of pollen from three transects across the slide. However, you then switch to pollen volume without giving a good idea of (a) how you calculate this (something about the scoring of pollen load size – also should note is full load or 1/5 load consistent in volume across species?) and (b) why this is a better metric than just % for pollen.

156: “both pollen types”? Did you mean “pollen from both of these plant genera”?

216-217: Detail what “remaining taxa combined as background” means. From Figure 1, I can’t really tell what “background” means (I thought perhaps outgroups, but maybe you mean the small clades?).

249-250: I have done relatively little modelling of this sort, but I would be somewhat concerned if the models are estimating negligible extinction rates – do you mean fairly equal across clades? Or, as written, that there has been basically no extinction? Or something else entirely? Clarify.

262-279: Pollen volume? From the methods, I thought you counted grains across three transects on a slide and used percentages of the total. Detail better in methods the conversion to volume.

344-345: Can you describe what instead they thought drove this pattern? Or was it simply untested? Either way, it should be noted.

347: “and in fact they are more frequent than previously thought”. Either a reference to say that these shifts are rare, or just delete the clause. Not sure it adds that much – you convincingly show shifts and I trust that the other references you cite do, as well.

351: I would avoid “Specialised” here and change it to “inaccessible” as you have elsewhere.

Specialised, to me, means relying on a single or few pollinators, which is not what you mean here, I believe. Nobody argues that *Trifolium* is specialized – though it is bee-pollinated, for sure!

352: I think you have the tribes wrong, and your references do not support your assertions. I thought *Trifolium* was *Trifoliae*? *Astragalus Galegae*? I know these are all closely related and I don’t have access to the Lewis book you cite, but the Sanderson and Wojciechowski paper has a bewildering array of tribes in there. Anazi (2019) the most recent reference that you cite does not make any mention of tribes, but cites Schaefer et al (2012), which says that *Fabeae* is just *Pisum*, *Lens*, *Vicia*, *Lathyrus* and *Vavilovia*. If they are monophyletic, note that, but I don’t think these are all *Fabeae*, and therefore this tribe certainly does not represent “the great majority of temperate herbaceous *Fabaceae*”.

363: “were” to “where”. It is worth noting that *Fabaceae* are abundant and widespread everywhere, and most diverse in the tropics, so more explaining might be helpful – i.e. *Trifolium*/other herbaceous fabs might be what you are specifically discussing?

372: Might be nice to note that you, too, recovered that association.

372-373: Why is the *Melissodes* preference important here? And how does it lead into the concluding sentence? *Xenoglossodes* is nestled in your clade (even if paraphyletic), *Melissodes* would be a distant outgroup and would say nothing more about *Xenoglossodes* than any of the other groups. I’d either explain the link you are trying to make here more clearly or delete.

373-376: Sort of the same as the previous comment, but why does radiation on *Asteraceae* for *Melissodes* (and the diversification of *Tetralonia* and *Xenoglossodes* not on *Fabeae*) set up the *Synhalonia* radiating with the switch to restricted pollen? Make this abundantly clear.

381-402: Your ACRs (presented in Figure 1 and tables S3) are about accessibility, not on plant family. Looking at Figure 1, there are many asters represented, but also other families of accessible pollen (i.e. cucurbits, malvs, mustards, etc.). You cannot say that they have “aster specialist ancestors” or that *Asteraceae* is the ancestral host (i.e. it looks like *Tetralonia* probably has aster as an ancestral state, but it seems unlikely that the other clades do).

412-417: This conclusion is a non-sequitur from the pollen chemistry focus of the paragraph. I think I might put it above the pollen chemistry (The flow would then be: Unobserved hidden traits had an effect.... One very likely factor would be pollen chemistry).

414-415: This point needs to be detailed a bit better, with references and some interpretation. This is why I asked for the sampling and total species numbers in the description of the system, since a lower sampling amount (and 20% is actually pretty good!!!) is a caveat, and not only to SSE models, but to all the interpretation – even ancestral states can be strongly influenced if sampling is low (and there may be this issue for the *Eucera* section, as you note it evolved in american deserts, yet your sampling is entirely palearctic for this group).

374/418/422: “advantage”/“advantageous” rephrase and be specific, i.e. higher diversification rates – I don’t think that a higher diversification rate is necessarily an advantage in an ecological sense (and seems nonsensical in an evolutionary sense with lots of switches).

424: delete “speciliases or” – it reads more clearly as “bee fauna largely depends on a few...”

426-430: This whole paragraph needs to be rewritten and rethought, as these references do not support your statements at all. The 2000 Minckley paper discusses specialization on *Larrea* – an

actinomorphic, open flower – and the 2013 paper discusses *Larrea* and *Prosopis* (an accessible, with pollen in the open, woody, fab). You do not mention exploitation of *Zygophyllaceae* at all and you make it seem like all *Fabaceae* hosts are herbaceous and with restricted pollen. Herbaceous species in deserts are not reliable and abundant (though they may be the latter in certain years), and your assertion that “association with these host plants were maintained because they offered abundant and reliable resources” is not at all supported, since the abundant and reliable host genera (i.e. those dominant desert shrubs) are not in your host list at all (creosote shows up once in your reported host list; mesquite not at all). Looking through the SI material, the one apparent desert species (e.g. *Synhalonia primiveris*) are primarily feeding on the exact opposite of this assertion – unreliable annuals (*S. primiveris* does feed from *Psorothamnus*, *Hyptis*, *Prunus*, *Purshia*, *Larrea* and *Tamarix*, but those are little of the pollen you recovered). Despite your assertion that *Eucera* species “evolved in the xeric habitats of North America” (L423) you sampled only Palearctic species. (I am assuming that you mean the *Eucera* clade, as indicated in 419, not the whole genus)

428-431: The abundant and reliable species are not the restricted ones, as implied.

436: Again, be more specific instead of “evolutionary success” – increased diversification rates.

Figure 2: I might put a line at 0, just to indicate where the no effect would be (and that your most of the probability distribution under both models is far above that point). I think that would make your result a bit more striking.

Review form: Reviewer 2

Recommendation

Major revision is needed (please make suggestions in comments)

Scientific importance: Is the manuscript an original and important contribution to its field?

Excellent

General interest: Is the paper of sufficient general interest?

Excellent

Quality of the paper: Is the overall quality of the paper suitable?

Good

Is the length of the paper justified?

Yes

Should the paper be seen by a specialist statistical reviewer?

Yes

Do you have any concerns about statistical analyses in this paper? If so, please specify them explicitly in your report.

No

It is a condition of publication that authors make their supporting data, code and materials available - either as supplementary material or hosted in an external repository. Please rate, if applicable, the supporting data on the following criteria.

Is it accessible?

Yes

Is it clear?

No

Is it adequate?

Yes

Do you have any ethical concerns with this paper?

No

Comments to the Author

Authors present a very interesting study on the evolution of a species diverse group of solitary bees (i.e. genus *Eucera*) in relation to their pollen diet. Overall the manuscript is well written, simple and easy to follow (but see some comments below). The references are accurate, the analyses are well done and the discussion is well balanced. This article will interest a broad audience of evolutionary biologists. I recommend its publication.

The major concern that I have on this study is the ambiguity of the definition of the shift of floral choices. When I was reading the introduction, I thought that the authors will present a study on host-plant shifts while *Eucera* species are keeping a restricted diet (i.e. shift from oligolecty to another oligolecty). For example, a study on shifts from Asteraceae pollen diet to Fabaceae pollen diet. However the shifts of diet in *Eucera* clade are also related sometimes to an increase of diet breadth (i.e. shift from oligolecty to polylecty). They are becoming more generalist. In this case, there are actually some studies already showing that extension of pollen diet increased diversification rate in Bees (Murrey et al. 2018). In an evolutionary and physiological point of view, these two kinds of shifts (specialism to another specialism, and specialism to generalism) are very different. So, authors should introduced better this concept and the study of Murrey et al. 2018. Moreover they should additionally analyze the impact of generalism in the diversification of the genus *Eucera*.

Minor comments:

- Line 32: Murrey et al. 2018 prove that generalism in bees increased speciation rate.
- Introduction: justify when you focus your study on pollen and not nectar.
- Line 88: pollen from flower with restricted access can have particular chemical composition (as presented in the discussion). This concept should be introduce, partly to justify why you focus your study on pollen.
- Line 91: evolutionary advantage of generalism? Evolutionary advantage to forage on flower with restricted access? Avoid competition? Have access to better reward?
- Line 95: how many species of *Eucera*?
- In the results section, you mention that you considered data from literature for floral observations. You don't mention it in the M&M section. How do you make the difference between data for nectar and pollen collection?
- Line 349: see also study on *Colletes* (Müller et Kuhlmann 2008)
- Line 360: Fabaceae
- Lines 346 - 417: I agree with this interesting section on secondary metabolites. However authors should also discuss about protein and lipid contain of Asteraceae versus Fabaceae. See the study of Vaudo et al. 2020.
- Line 384: there are a lot of study on the development of bumblebees on different diet (e.g. Vanderplanck et al. 2017; Tasei & Aupinel 2008; Moerman et al. 2016)
- Line 380: I would cite and discuss the study of Vanderplanck et al. 2017 (and not 2018).
- Figure 1: I was expecting to see the information R versus A on the figure. It would be usefull to add this information, while the code of the specimen is not really important to mention.
- SI: host of *Synhalonia acerba*, restricted or accessible? I would say that access to Ericaceae pollen is restricted.

Additional references (not cited by the authors)

Vaudo A., Tooker J., Patch H., Biddinger D., Coccia M., Crone M., Fiely M., Francis J., Hines H., Hodges M., Jackson S., Michez D., Mu J., Russo L., Safari M., Treanore E., Vanderplanck M., Yip E., Leonard A., Grozinger C. 2020. Pollen protein: Lipid macronutrient ratios may guide broad patterns of bee species floral preferences. *Insects*, 11(2): 132.

Vanderplanck M., Vereecken N.J., Grumiau L., Esposito F., Lognay G., Wattiez R., Michez D.

2017. The importance of pollen chemistry in evolutionary host shifts of bees. *Scientific Reports*, 7 : 43058.

Moerman M., Roger N., De Jonghe R., Michez D., Vanderplanck M. 2016. Interspecific variation in bumblebee performance on pollen diet: new insights for mitigation strategies. *Plos One*, 11(12): e0168462

Decision letter (RSPB-2020-2003.R0)

20-Oct-2020

Dear Dr Dorchin:

I am writing to inform you that your manuscript RSPB-2020-2003 entitled "Bee-flowers and bees, evidence for interaction-driven diversification" has, in its current form, been rejected for publication in *Proceedings B*.

This action has been taken on the advice of referees, who have recommended that substantial revisions are necessary. With this in mind we would be happy to consider a resubmission, provided the comments of the referees are fully addressed. However please note that this is not a provisional acceptance.

Sincerely,
Dr Locke Rowe
mailto: proceedingsb@royalsociety.org

Associate Editor
Board Member: 1
Comments to Author:

In this paper the authors investigate a biologically very interesting question, whether the interaction between plants and pollinators might have shaped the pollinator's macroevolutionary

dynamics, in this specific case the bee genus *Eucera* (Apidae). It collects an impressive data set, uses the most recent phylogeny and do take into account several methodological aspects. That is why I was initially very excited about the paper, but after reading the paper a second time, I think there are some important methodological aspects (see my comments below) that could impar the most relevant result. Reviewers raised some really relevant aspects that could be addressed but the methodological aspect I will briefly explain will require a reanalysis of the data and no guarantee that the results will remain the same. My gut feeling is that the results will change (no evidence for differential diversification rate, or at least not able to reject the hidden state), but I could be wrong.

My main concerns relate to some the “side effects” of having a smaller sampled size (about 20% of the bee species are sampled) when doing diversification analysis. Sometimes (most time) this is the reality of the data, and in fact the authors have gathered an impressive dataset on pollen, so I have to admit that I am thorned to be criticizing this aspect, but as I will explain I suspect that the results might have emerged from those “side effects” and some methodological choices. I commend the authors for taking several steps to mitigate and explicitly deal with these “side effects” (e.g. incorporate the sampling fraction on BiSSE and HiSSE) but on a second thought I think there are still some relevant aspects. I will try to explain my reasoning here and I hope this helps the authors in future submissions of this manuscript.

1) In the methods section the authors say they used the diversitree function “make.bisse.uneven” to inform the analysis what is the percentage of missing species associated to each subgenera. If I get this right, the BiSSE analysis attributes this percentage of missing species to the crown age of the clade of interest. The problem with this is that under-sampling is likely to affect the age estimate of the crown group itself, and hence the posterior estimates of rate. This is particularly likely when the under-sampling is high, which is the case for several subgenera here. Hence differences in rates might be strongly related to differences in how far each lineage is from the “true” crown group age and how well sampled is each subgenera. This is ameliorated by the very clever idea to use the same frequency of equal states for the missing species within each genus, but it is still possible that an effect is relevant here.

2) More importantly, when doing the HiSSE analysis, which is extremely relevant to discard the potential effect of a hidden state driving the differences detected on the BiSSE analysis, the sampling fraction of missing species used was not for each subgenera (as for the BiSSE), but for the “whole phylogeny”, which is not a problem per se (in fact I think this is the only automatic option for HiSSE), but there is one potential problem in the context of the paper. Given that about 80% of species are missing, it is possible that the choice of the BiSSE model over the HiSSE model was somehow affected by differences in how the sampling fraction was implemented on the phylogeny. This is because in the hidden state model most of the potential rate heterogeneity was inserted as “white noise” in a sense (80% of species added with equal frequencies for each state throughout the whole tree). Hence it is possible that a better fit for the BiSSE model (which is the key result to suggest that the variable of interest, the type of pollination, affects diversification rates) results from this difference on how to handle the missing data. This effect might be exacerbated by the potential effect mentioned above on point “1”.

SUGGESTION: A better comparison would be to compare the HiSSE model to the BiSSE model where the sampling fraction is not informed clade by clade but for the whole phylogeny in both analyses. In this way the comparison in more adequate and the potential problem associated to the inclusion of species in the crown age disappears. My intuition is that this will remove the statistical difference between HiSSE and BiSSE, but I could be wrong. Alternatively, if the authors devise a way to include the missing species through the whole stem lineage (not only through in the crown group) and do that for both for the HiSSE and BiSSE analyses, I think this might be another way to more appropriately compare the results and rule out the effect of a hidden state.

Additional comments/suggestions are:

A) A bit more information about the lineage (could be very short) before the discussion would help the reader visualize the study/sampling methods (this was also mentioned by one reviewer). For example, where is the genus geographically found? Is the pollen data derived from specimens collected through the whole genus distribution or for only parts of it? Some of that information might be in the supplemental material but a few sentences in the main text might be helpful.

B) Would it be interesting to try a fourth character data set where species classified as “Accessible + restricted” were considered “Accessible”, and the species classified as “restricted” was scored as R in the data matrix only if had restricted? In the first character data set the authors scores A if species has only A, and R if species has A + R right? I understand the reasoning here (which could be made explicit in the main text), but I wonder if a fourth scenario (doing the opposite scoring scheme) would not help reinforce that the R state is what is driving the increase in diversification rate. My concern is that the higher rate of “R” might be influenced by the rates experienced by those species that have both R + A. Here it would behave as a hidden state. I know the HiSSE model was designed for that but given that under-sampling was treated differently (see comment above), this would be another potentially interesting test. Doing this alternative scoring might not be that relevant (in particular in the light of the second score matrix where some species are scored as ambiguous), but an explicit explanation why R + A was coded as R in the first data matrix might help the reader.

C) This is a minor fix, but I was initially a bit confused on how the MCMC result on diversification rate was evaluated. When I got to figure 2 I understood it better but a better explanation might be necessary at the methods. For example, it is said (lines 206-208) that “The posterior probability that state R exhibits higher net diversification rate than state A is represented by the percentage of MCMC steps in which $dR > dA$.” This gave me the initial impression that is the frequency of times the MCMC finds a higher values for dR rather than the magnitude of the difference in $dR-dA$ (shown in figure 3) that was used as evidence in favor of $dR > dA$. I guess both aspects were used (frequency and magnitude) but somehow reading the text I felt that the magnitude aspect was not explicitly presented. Just make it more explicit in the methods section. E.g. you will plot the posterior probability of differences as shown on figure 2

Given the issues raised above, and those raised by the reviewers, I unfortunately cannot recommend the paper for publication in its current form. If my impressions regarding the differential treatment of missing species (BiSSE vs HiSSE) is correct, it will require a reanalysis of the data which might substantially change the results. Do not get me wrong, I think the work is interesting, the data set amazing, and it should be published, but depending on the new results, perhaps in a more specialized journal and/or focusing on other aspects rather than the diversification rates. If the authors think they can properly address those issues and the main result are still interesting to a broad audience, then a resubmission would be welcomed. Given the uncertainty of the re-analysis and that the results (or focus of the paper) might substantially change, I think this potential resubmission should be treated as a new submission.

Reviewer(s)' Comments to Author:

Referee: 1

Comments to the Author(s)

Review of Dorchin et al.

This study is great effort to get at a question which ought to be interesting to a great deal of researchers. As a person strongly on the plant-side of pollination, I was unaware that there was little work on diversification of bees in response to floral morphology. The pollen data which they assembled was surely a great deal of work and the choice of a group of bees and the sampling (20% of the clade) is quite high for these larger-scale ecological comparative analysis. I think this paper is certainly of the suitable for publication in Proc B after revisions and I am sure it will be of interest to many. I have pages of individual comments, but they broadly fall into:

- 1) Clarity issues – there are a lot of methods and results which are not explained sufficiently to understand exactly why you chose to do them. For instance, why is pollen volume used and how is it calculated from the percentages gathered? And if the bee was collected early in a foraging bout, why should a lower volume of pollen be any less indicative of preference?
- 2) There are a few long paragraphs with multiple ideas that need to be shortened and clarified (i.e. 376-417).
- 3) Many references are needed throughout for various statements and in some cases, references should be checked for accuracy (i.e. final paragraph).
- 4) Certain word choice is vague and needs to be made much clearer: e.g. “Evolutionary success” is used to indicate an increase in diversification rates.
- 5) The final paragraph about the evolution of the specialist group and the flowers they rely on needs to be significantly rethought and re-referenced.

Title: It seems like the novelty of this study is that you focus on the bees, yet the title says “bee flowers” first, and until I read the abstract, I thought this would be a plant paper. I’d rephrase somehow to stress the bee aspect. (Even, perhaps, just reversing bees and bee flowers).

30-31: I’d be a little more circumspect here (and elsewhere) with word choice – you don’t really know that these bees (or any floral visitor!) is truly a pollinator, or if a pollinator, a good one or one important to the plant. You are very careful with this elsewhere (i.e. the nice system/study description in L95-115). It is worth being very specific, even in these broad sections like L30-34, as the bees needn’t be pollinators to have diversifying in response to floral preference/floral traits. “Floral visitor” is the phrase I use most often, but that’s not really appropriate here – perhaps “pollen consumer” or something similar. Loads of even specialist bees/moths/etc, are not particularly good pollinators, but are reliant on one or few plants.

55-56: The clause “and many floral traits are used in the characterization of angiosperm species” is a non sequitur here, and interrupts the flow between pollination ideas. I’d delete or rephrase this sentence.

57: What exactly is the assumption here? That pollinators shape floral evolution? I think deleting the clause will make this clearer, but it might be worth rephrasing this section.

56-65: You definitely don’t have to take this suggestion, but in the two sentences before the concluding one, you discuss a lot of pollinator specialization (which implies some evolutionary knowledge), then say that there is little known about the evolution of pollinators. As I understand the literature, the diversification of plants after developing things like nectar spurs or long tubes is not actually demonstrated from pollinator specialization (i.e. most hummingbirds and hawkmoths are not specialists, in the sense that they hit a single species or genus of flowers), instead from these plant morphological changes decreasing the generality of their flowers; i.e. the plant is specializing, not really the pollinator (which you get at in L58-59), but then you switch to pollinator specialization and repro isolation in the next, and conclude that little is known about evolution of pollinators.

76-78: Might be nice to have a reference here: bees could certainly exploit flowers for both pollen and nectar without being long-tongued or co-evolving with the flowers (lots of native bees here nectar and collect pollen of non-native flowers, including restricted zygomorphic species, with which they have no co-evolutionary history at all).

81-83: This sentence about exceptions in bees, needs context, since it follows a plant sentence.

What exact exceptions do you mean? What flowers they exploit? The bee morphology (i.e. non-LT bees exploiting zygomorphic flowers)?

94: “potential evolutionary advantage” What do you mean by this statement? Later on, in your results, you use this to mean increased diversification, but this is distinctly not what you mean here. Rephrase or define exactly.

95-115: This is a great overview, but can you detail a little bit of their interaction (i.e. they collect pollen, do they also drink nectar? You collect pollen from the scopa later, but you mention

nototriby in the intro – do they also get pollen in other parts of the body or do they groom it down right away?). This is probably all known by most bee people, but this paper should attract attention of a lot of plant people, too, so a little more natural history of the bees would be nice for those of us who don't know bees as well.

102: Boraginaceae are actinomorphic and often shallow. I know that some are deep and specialized, but many readers will probably picture something like *Heliotropus* or *Nama*, which are actinomorphic and probably easily exploited by many floral visitors.

119: How many species total are there in *Eucera*? For the 85 species, how much of the radiation do they cover? This is at first glance a good sample size for comparative analyses (especially that you sampled pollen from 76 of the 85 species!!!), but it is worth taking the cautions detailed in Losos (*American Naturalist*, 2011, 177: 709-727) into consideration: i.e. if those 85 are drawn from across a phylogeny of 1500 species, there are some caveats to that analysis (in L95-96, you just write “one of the largest”, and as a plant person, I have no idea if this is 100, 500, or 1000 and that surely matters to interpretation). Later on you clarify that it is 20%, but this needs to be noted here, too.

131-134: What a dataset!

139: Why are percentages per species summed? I do not think you can sum percentages (i.e. 50% + 75% is uninterpretable); did you mean that they were they averaged? If that is what you mean, is this appropriate without some sort of weighting (i.e. if you had 1000 pollen grains on one bee, should that be averaged with another bee of the species which had 10 pollen grains?)

138-145 (and again 262-279): It seems as if you are calculating percentages of pollen from three transects across the slide. However, you then switch to pollen volume without giving a good idea of (a) how you calculate this (something about the scoring of pollen load size – also should note is full load or 1/5 load consistent in volume across species?) and (b) why this is a better metric than just % for pollen.

156: “both pollen types”? Did you mean “pollen from both of these plant genera”?

216-217: Detail what “remaining taxa combined as background” means. From Figure 1, I can't really tell what “background” means (I thought perhaps outgroups, but maybe you mean the small clades?).

249-250: I have done relatively little modelling of this sort, but I would be somewhat concerned if the models are estimating negligible extinction rates – do you mean fairly equal across clades? Or, as written, that there has been basically no extinction? Or something else entirely? Clarify.

262-279: Pollen volume? From the methods, I thought you counted grains across three transects on a slide and used percentages of the total. Detail better in methods the conversion to volume.

344-345: Can you describe what instead they thought drove this pattern? Or was it simply untested? Either way, it should be noted.

347: “and in fact they are more frequent than previously thought”. Either a reference to say that these shifts are rare, or just delete the clause. Not sure it adds that much – you convincingly show shifts and I trust that the other references you cite do, as well.

351: I would avoid “Specialised” here and change it to “inaccessible” as you have elsewhere.

Specialised, to me, means relying on a single or few pollinators, which is not what you mean here, I believe. Nobody argues that *Trifolium* is specialized – though it is bee-pollinated, for sure!

352: I think you have the tribes wrong, and your references do not support your assertions. I thought *Trifolium* was *Trifoliae*? *Astragalus Galegae*? I know these are all closely related and I don't have access to the Lewis book you cite, but the Sanderson and Wojciechowski paper has a bewildering array of tribes in there. Anazi (2019) the most recent reference that you cite does not make any mention of tribes, but cites Schaefer et al (2012), which says that *Fabeae* is just *Pisum*, *Lens*, *Vicia*, *Lathyrus* and *Vavilovia*. If they are monophyletic, note that, but I don't think these are all *Fabeae*, and therefore this tribe certainly does not represent “the great majority of temperate herbaceous *Fabaceae*”.

363: “were” to “where”. It is worth noting that *Fabaceae* are abundant and widespread everywhere, and most diverse in the tropics, so more explaining might be helpful – i.e. *Trifolium*/other herbaceous fabs might be what you are specifically discussing?

372: Might be nice to note that you, too, recovered that association.

372-373: Why is the *Melissodes* preference important here? And how does it lead into the concluding sentence? *Xenoglossodes* is nested in your clade (even if paraphyletic), *Melissodes*

would be a distant outgroup and would say nothing more about Xenoglossodes than any of the other groups. I'd either explain the link you are trying to make here more clearly or delete.

373-376: Sort of the same as the previous comment, but why does radiation on Asteraceae for Melissodes (and the diversification of Tetralonia and Xenoglossodes not on Fabaceae) set up the Synhalonia radiating with the switch to restricted pollen? Make this abundantly clear.

381-402: Your ACRs (presented in Figure 1 and tables S3) are about accessibility, not on plant family. Looking at Figure 1, there are many asters represented, but also other families of accessible pollen (i.e. cucurbits, malvs, mustards, etc.). You cannot say that they have "aster specialist ancestors" or that Asteraceae is the ancestral host (i.e. it looks like Tetralonia probably has aster as an ancestral state, but it seems unlikely that the other clades do).

412-417: This conclusion is a non-sequitur from the pollen chemistry focus of the paragraph. I think I might put it above the pollen chemistry (The flow would then be: Unobserved hidden traits had an effect.... One very likely factor would be pollen chemistry).

414-415: This point needs to be detailed a bit better, with references and some interpretation. This is why I asked for the sampling and total species numbers in the description of the system, since a lower sampling amount (and 20% is actually pretty good!!!) is a caveat, and not only to SSE models, but to all the interpretation – even ancestral states can be strongly influenced if sampling is low (and there may be this issue for the Eucera section, as you note it evolved in american deserts, yet your sampling is entirely palearctic for this group).

374/418/422: "advantage"/"advantageous" rephrase and be specific, i.e. higher diversification rates – I don't think that a higher diversification rate is necessarily an advantage in an ecological sense (and seems nonsensical in an evolutionary sense with lots of switches).

424: delete "speciliasies or" – it reads more clearly as "bee fauna largely depends on a few..."

426-430: This whole paragraph needs to be rewritten and rethought, as these references do not support your statements at all. The 2000 Minckley paper discusses specialization on Larrea – an actinomorphic, open flower – and the 2013 paper discusses Larrea and Prosopis (an accessible, with pollen in the open, woody, fab). You do not mention exploitation of Zygophyllaceae at all and you make it seem like all Fabaceae hosts are herbaceous and with restricted pollen.

Herbaceous species in deserts are not reliable and abundant (though they may be the latter in certain years), and your assertion that "association with these host plants were maintained because they offered abundant and reliable resources" is not at all supported, since the abundant and reliable host genera (i.e. those dominant desert shrubs) are not in your host list at all (creosote shows up once in your reported host list; mesquite not at all).

Looking through the SI material, the one apparent desert species (e.g. Synhalonia primiveris) are primarily feeding on the exact opposite of this assertion – unreliable annuals (S. primiveris does feed from Psorothamnus, Hyptis, Prunus, Purshia, Larrea and Tamarix, but those are little of the pollen you recovered). Despite your assertion that Eucera species "evolved in the xeric habitats of North America" (L423) you sampled only Palearctic species. (I am assuming that you mean the Eucera clade, as indicated in 419, not the whole genus)

428-431: The abundant and reliable species are not the restricted ones, as implied.

436: Again, be more specific instead of "evolutionary success" – increased diversification rates.

Figure 2: I might put a line at 0, just to indicate where the no effect would be (and that your most of the probability distribution under both models is far above that point). I think that would make your result a bit more striking.

Referee: 2

Comments to the Author(s)

Authors present a very interesting study on the evolution of a species diverse group of solitary bees (i.e. genus Eucera) in relation to their pollen diet. Overall the manuscript is well written, simple and easy to follow (but see some comments below). The references are accurate, the analyses are well done and the discussion is well balanced. This article will interest a broad audience of evolutionary biologists. I recommend its publication.

The major concern that I have on this study is the ambiguity of the definition of the shift of floral choices. When I was reading the introduction, I thought that the authors will present a study on host-plant shifts while Eucera species are keeping a restricted diet (i.e. shift from oligolecty to

another oligolecty). For example, a study on shifts from Asteraceae pollen diet to Fabaceae pollen diet. However the shifts of diet in Eucera clade are also related sometimes to an increase of diet breadth (i.e. shift from oligolecty to polylecty). They are becoming more generalist. In this case, there are actually some studies already showing that extension of pollen diet increased diversification rate in Bees (Murrey et al. 2018). In an evolutionary and physiological point of view, these two kinds of shifts (specialism to another specialism, and specialism to generalism) are very different. So, authors should introduced better this concept and the study of Murrey et al. 2018. Moreover they should additionally analyze the impact of generalism in the diversification of the genus Eucera.

Minor comments:

- Line 32: Murrey et al. 2018 prove that generalism in bees increased speciation rate.
- Introduction: justify when you focus your study on pollen and not nectar.
- Line 88: pollen from flower with restricted access can have particular chemical composition (as presented in the discussion). This concept should be introduce, partly to justify why you focus your study on pollen.
- Line 91: evolutionary advantage of generalism? Evolutionary advantage to forage on flower with restricted access? Avoid competition? Have access to better reward?
- Line 95: how many species of Eucera?
- In the results section, you mention that you considered data from literature for floral observations. You don't mention it in the M&M section. How do you make the difference between data for nectar and pollen collection?
- Line 349: see also study on Colletes (Müller et Kuhlmann 2008)
- Line 360: Fabaceae
- Lines 346 – 417: I agree with this interesting section on secondary metabolites. However authors should also discuss about protein and lipid contain of Asteraceae versus Fabaceae. See the study of Vaudo et al. 2020.
- Line 384: there are a lot of study on the development of bumblebees on different diet (e.g. Vanderplanck et al. 2017; Tasei & Aupinel 2008; Moerman et al. 2016)
- Line 380: I would cite and discuss the study of Vanderplanck et al. 2017 (and not 2018).
- Figure 1: I was expecting to see the information R versus A on the figure. It would be useful to add this information, while the code of the specimen is not really important to mention.
- SI: host of *Synhalonia acerba*, restricted or accessible? I would say that access to Ericaceae pollen is restricted.

Additional references (not cited by the authors)

- Vaudo A., Tooker J., Patch H., Biddinger D., Coccia M., Crone M., Fiely M., Francis J., Hines H., Hodges M., Jackson S., Michez D., Mu J., Russo L., Safari M., Treanore E., Vanderplanck M., Yip E., Leonard A., Grozinger C. 2020. Pollen protein: Lipid macronutrient ratios may guide broad patterns of bee species floral preferences. *Insects*, 11(2): 132.
- Vanderplanck M., Vereecken N.J., Grumiau L., Esposito F., Lognay G., Wattiez R., Michez D. 2017. The importance of pollen chemistry in evolutionary host shifts of bees. *Scientific Reports*, 7 : 43058.
- Moerman M., Roger N., De Jonghe R., Michez D., Vanderplanck M. 2016. Interspecific variation in bumblebee performance on pollen diet: new insights for mitigation strategies. *Plos One*, 11(12): e0168462

Author's Response to Decision Letter for (RSPB-2020-2003.R0)

See Appendix A.

RSPB-2021-0533.R0

Review form: Reviewer 1

Recommendation

Major revision is needed (please make suggestions in comments)

Scientific importance: Is the manuscript an original and important contribution to its field?

Excellent

General interest: Is the paper of sufficient general interest?

Excellent

Quality of the paper: Is the overall quality of the paper suitable?

Marginal

Is the length of the paper justified?

Yes

Should the paper be seen by a specialist statistical reviewer?

No

Do you have any concerns about statistical analyses in this paper? If so, please specify them explicitly in your report.

No

It is a condition of publication that authors make their supporting data, code and materials available - either as supplementary material or hosted in an external repository. Please rate, if applicable, the supporting data on the following criteria.

Is it accessible?

Yes

Is it clear?

No

Is it adequate?

Yes

Do you have any ethical concerns with this paper?

No

Comments to the Author

I previously reviewed this manuscript (reviewer 1), and feel that the author's responses to my queries were largely satisfactory. I certainly defer to the editor's judgement for diversification analyses. I think the authors' inclusion of a few extra groups in the phylogeny (and not pollen data set) did little to address my concerns about completeness or ACR, but given that this is the data set that needs to be worked with – and is impressive – I do not see a real problem there. It is certainly easy to imagine that that the 80% of non-sampled species could shift, or even reverse, some of the conclusions. But that's how science works and is not a failing of the paper.

This revision, I delved into the plants, as that is where my expertise is most germane. The biggest issue is that the classifications are completely opaque – its not in the methods anywhere. This time I read through the supplement much more in depth – but not with a fine-toothed comb - and my biggest complaint is that you need to really describe how you came up with the

restricted/accessible, oligolectic/polylectic classifications; they don't really align with what I would score from the data. There are a bunch where there were many known host families, AND you recovered multiple pollen, yet you classified them as oligolectic. Others where the fabaceous genera listed were not restricted.

I think you need to either assign uncertainty in those classifications (and bootstrap) or come up with some really strict guidelines that make sense. These will, absolutely, change the classification of some species, so the analyses will need to be rerun.

My other big issue is still lack of specific language.

2 - Title: I still disagree with the inclusion of flowers; there is no evidence that they diversify from bees in this paper. This title does not accurately represent the content of the paper.

31: In your response to my previous comments, you say that you changed pollinator to floral visitor throughout. You did not; it is used incorrectly here. (i.e. in 31, you really mean - insects which utilize floral resources - pollination has nothing to do with the diversification, your paper is about floral resources). The preceding two are accurate, as you are focusing on the plant perspective.

34: Is it truly preference? Or just use/utilization? You don't have data on abundances to say they are preferentially using any species, just that they are using them (possibly exactly in line with abundance).

63: pollinators, again -> "those insects which utilize floral resources"

75: Why did they need to coevolve? I think you mean "evolved" here.

77: Bilaterally symmetric zygomorphic flowers need not have restricted pollen, nor does concurrent diversification imply coevolution. Change the sentence before and this will be more accurate.

97-98: "While all species can potentially extract nectar from bee-flowers" What does this mean? There are lots of generalized flowers that they can presumably get nectar from, too. "different sources of pollen" means specifically those plants that they do not nectar on? This section needs revision.

99-105: You are talking only about pollen collection here, right? Make that clearer before L 106.

105-106: "pollen specialists" belongs after "oligolectic species", as it defines it (same with generalists).

109: What trait? Specialization on a single family? Or accessing restricted pollen? Ambiguous as written.

111-112: "laborious work of palynological expertise". Nonsense as written. Do you mean "laborious work AND palynological expertise".

157: When you say only restricted pollen was recorded, do you mean you left out those species without restricted pollen when encountered or they never showed up in your dataset? They show up in your appendices as hosts, so you need to be clear here. (i.e. you have *Petalostemon*, *Dalea* as "restricted hosts" of *Xenoglossodes albata* - that's not correct at all - though *Psorothamnus* is - since you observed them on *Dalea*, how did you differentiate that 55% fab pollen into restricted? Plus the asters are all open).

257/258: Why pollen preference? Not pollen host or something else. Preference has a very distinct definition in foraging and that is not what you are using here.

269: Its not 36 early-diverging lineages - its 36 species from early diverging lineages, right?

272: Zygothyllaceae spelled wrong

275: there are not 44 early diverging subgenera.

288: "with high probability" belongs after inferred.

291-295: "preference to" -> "utilization of" (both the definition of preference, but also awkward phrasing).

331: not foraging behaviors - resource use.

332: delete comma

332-336: I don't understand the zygomorphy/actinomorphy part here. You are specifically talking about Fabaceae as the restricted/zygomorphic host, as borages are actinomorphic/restricted. I guess I think the symmetry is confounded with the fab/aster comparison (if Lamiaceae or Plantaginaceae were really big in the analyses, then sure, you could make a broader conclusion about symmetry, but they aren't).

338: morphological similarity of the bees or plants? Awkward phrasing.

346: is “higher Megachlidae” a normal phrase? I think evo. biologists (or at least botanists) are trying to get away from using higher/lower in favor of more specific/accurate phrasing (few say higher plants anymore).

375: They are “early-diverging” but since the species you sampled are modern, you can’t really say they are basal.

385: pollen is known as a poor source of pollen? You mean nutrition?

388: Delete “Especially”.

403: delete “unfavourable”

411: smaller than exactly what?

Figure 1: italicize specific names.

Supplement:

I didn’t really want to go through this and I just skimmed it, but I guess as a plant person, I should note some that make no sense to me. They make no reasonable distinction between polylectic and oligolectic, there are botanical errors, etc. I think they should be changed or defended and possibly the analyses rerun. Honestly, you need to proof the list and maybe look at specimens of some genera to determine accessible/restricted. There are some genera that are considered restricted in one species and accessible in another (i.e. *Dalea* in *X. albata*/*X. eriocarpi*). Also some genera are assigned to the wrong family (again, *Dalea* in *X. albata*).

If your definition of oligolectic is a single family, as is stated in L. 105-106, the assignments don’t make sense. Your actual criteria needs to be stated (see my broad comments above).

P. venusta/*P. carinata* – why is the former polylectic and the latter oligolectic? Both have the same a single family from pollen but more known host families.

P. carinata – lowercase specific epithet.

T. labrosa/*nigripilosa* – how is this oligolectic? 17% of pollen was from other families, and other families listed in the literature.

T. nigriceps – 40% of pollen is not from asters. This is not oligolectic.

T. tenuifasciata – again, how oligolectic?

X. eriocarpi – how is something with known hosts across 10 families oligolectic?

Xenoglossodes albata: This species is classified incorrectly as “restricted pollen” as most of the genera have their anthers out in the open, both the fabs (*Petalostemon*/*Dalea* – possibly referring the same species) and the asters (*Amorpha*, *Echinacea*) as “restricted hosts” of– that’s not correct at all – though *Psoralea* is – since you observed them on *Dalea*, how did you differentiate that 55% fab pollen into restricted? Also, *Dalea* is listed as both an aster and a fab! It’s a fab.

X. lippiae – again why oligolectic?

Synhalonia actuosa has *Zygophyllaceae* listed as pollen, but not as a host.

Synhalonia frater – *Hydrophyllum* is a borage, not a fab (and it is open, not restricted). There are no *Ericaceae* listed.

Synhalonia hamata – Some of the borages listed are open, as are the plantages (sure they have tubes, but the pollen is accessible). Also, with that host list, how is it considered oligolectic?!? You define that in the introduction as a single botanical family; even the pollen taken off is from three families. Sure, those two fell below the 5% threshold each, but together the other families were 9%.

Eucera dafnii – Neither restricted or oligolectic make sense with the entries. The only two genera listed are nonrestricted. Its not clear what the fab pollen came from, but I think you cannot say its oligolectic, even though the one sample had high fab pollen (also, not clear if restricted or not without a genus).

E. proxima – again, why oligolectic? *Anchusa* is accessible, I’ve no experience with *Hormuzakia*.

E. nigrilabris – again, why oligolectic? Looks like a lot of hosts.

E. clypeata – why oligolectic/restricted? *Caryophyllaceae* is almost certainly accessible (I think even many of tubed *Silene* have the anthers everted)

Review form: Reviewer 2

Recommendation

Accept with minor revision (please list in comments)

Scientific importance: Is the manuscript an original and important contribution to its field?

Excellent

General interest: Is the paper of sufficient general interest?

Excellent

Quality of the paper: Is the overall quality of the paper suitable?

Excellent

Is the length of the paper justified?

Yes

Should the paper be seen by a specialist statistical reviewer?

Yes

Do you have any concerns about statistical analyses in this paper? If so, please specify them explicitly in your report.

No

It is a condition of publication that authors make their supporting data, code and materials available - either as supplementary material or hosted in an external repository. Please rate, if applicable, the supporting data on the following criteria.

Is it accessible?

Yes

Is it clear?

Yes

Is it adequate?

Yes

Do you have any ethical concerns with this paper?

No

Comments to the Author

Authors did a really good job in answering to all comments / recommendations from the referees and editor. I have one last recommendation but this is not critical (that's why I proposed a minor revision). As authors considered now the level of diet specialisation in their analyses of diversification rate (great!), I think that they should do the same for the analyses of Ancestral state reconstruction. They actually already discussed about potential results in the Discussion section (line 373).

Minor comments:

- Line 95: You should add a sentence to justify why you selected Eucera to explore this topic.
- Line 157: I did not understand how you recorded Fabaceae. You mean, all Fabaceae have restricted pollen type? Or you did not record any non-restricted Fabaceae pollen (like from Mimosa).
- Line 211: typo on performed

- Lines 269 and 275: I would say "36 species from early diverging lineages" and "44 species from recently diverging subgenera".
- Line 343: I did not understand the last part of the sentence.

Decision letter (RSPB-2021-0533.R0)

06-May-2021

Dear Dr Dorchin:

Your manuscript has now been peer reviewed and the reviews have been assessed by an Associate Editor. The reviewers' comments (not including confidential comments to the Editor) and the comments from the Associate Editor are included at the end of this email for your reference. As you will see, the reviewers and the Editors have raised some concerns with your manuscript and we would like to invite you to revise your manuscript to address them.

Research ethics:

Use of animals and field studies:

It is a condition of publication that you make available the data and research materials supporting the results in the article (<https://royalsociety.org/journals/authors/author-guidelines/#data>). Datasets should be deposited in an appropriate publicly available repository and details of the associated accession number, link or DOI to the datasets must be included in the Data Accessibility section of the article (<https://royalsociety.org/journals/ethics-policies/data-sharing-mining/>). Reference(s) to datasets should also be included in the reference list of the article with DOIs (where available).

Please submit a copy of your revised paper within three weeks. If we do not hear from you within this time your manuscript will be rejected. If you are unable to meet this deadline please let us know as soon as possible, as we may be able to grant a short extension.

Best wishes,

Dr Locke Rowe

Associate Editor

Comments to Author:

I commend the authors for their hard work to answer all the concerns raised by both reviewers and me. I think this is much improved version, in particular with respect to explaining some of the procedures. That said, I still think there are two aspects that need attention.

The first relates to the under-sampling and how that level of under-sampling could generate biases. The authors have now explained better their procedure and indeed some of my previous suggestions were already implemented. That said, given the level of under-sampling, and in particular our limited knowledge on how those models might misbehave in a myriad of

situations that we have not fully explored, I think that a little bit of extra work needs to be done. I say this because the main result of the paper (difference in rates associated to different types of pollen) strongly rely on a statistical test where only about 20% of the species were used. Sure, the authors have taken some precautions/recommend procedures but given the level of under-sampling I think it would be necessary to run a few simulations to specifically investigate potential biases. Please see below some thoughts on that respect.

The second aspect was now raised by one of the reviewers and relates to the classification of pollen. According to the reviewer, who is a plant expert, the authors need to better describe how they came up with the restricted/accessible, oligolectic/polylectic classifications". Reviewer 1 presents a series of queries on the classification of individual specie which he felt were not classified correctly. Given that this is crucial to the main analysis, I think a better explanation of how this was done is warranted. Maybe a comparison/validation with primary literature (there must be some host plant use lists) would be a way to access the scoring of some dubious records. In that respect, a sensitive analysis (treating those dubious records differently) might be interesting, especially given the low sampling of species in the phylogeny. Few mis assignments might even change the result outcome.

Here I present a few thoughts and comments on the under-sampling and diversification methods that I hope will be helpful:

1) Regarding the placement of the missing species, as far as I understand there are two procedures described by Fitzjohn et al 2009. The first one they refer to "skeletal tree" and the second "terminally unresolved tree". In the first case there is a backbone tree fully resolved but incomplete with respect to sampling. Here the missing species are place within this tree. In the second case the tree "contain all species, but their relationships are not fully resolved, with some species grouped into unresolved clades". In the case of the "terminally unresolved tree", for those "unresolved clades" there is, by definition, no crown age, because all species are collapsed into the sub-clade stem age. In that sense, yes, it can consider the possibility of missing species to be placed from the stem age because such clade lacks structure. I understood (and I could be wrong) that in your case, you had a tree with structure (fully bifurcating) where several sub-clades have more than one species, and hence it is possible to estimate a crown age for that sub-lineage. If you were using "terminally unresolved tree", then yes, by definition it is the stem age and my comment of the effect of age of crown group does not make sense (but see further comment below). On the other hand if you are using a "skeletal tree" fully resolved, I am not sure exactly how the function "make.bisse.uneven" (which I presume might have been used) allows for defining either the stem age or crown age. I took a look at the function and it seems that indeed you might be able to choose what to use (stem or crown age). Hence the authors are correct to say that the missing species can be "included" using the stem age of a defined sub-clade (node). In that respect my comment on the under-sampling affecting crown age estimates, and hence rates estimate, was indeed not fully correct. Given my misunderstanding regarding some of those issues, I suggest the authors to provide the scripts used for the diversification analysis. Apart from being very useful for reviewers and editors to understand what was exactly done, it might be useful for others in the future when trying to set up similar analysis.

2) Although the authors are right with respect to the possibility of considering the stem age when allowing for under-sampling to be taken care, I still think that under sampling might affect rate estimates in a similar fashion because under-sampling should also affect the estimate of stem age, in particular if a missing species "intrudes" between two clades and if it that species does not belong to (was considered in the sampling fraction of) that lineages. So, although I might have been indeed incorrect to assume that diversitree forces the placement of missing species to be placed after the crown age, the same reasoning might still apply. That should be less problematic (although in practice we do not know to what extent) if the tree has all major sub-groups (e.g. all genera), but we would have to heavily rely on taxonomy to think that such problem is not relevant if we have all the sub-groups represented. I am not a bee expert so I cannot judge how the subgroups were represented, or even if we can reliably have a sense in this particular case what would be those sub-groups, how to assign the missing species to each one (I guess based on

taxonomy, but...), but an under-sampling where about 80 % are missing makes me suspicious we will not run in this problem. That said you mentioned that representatives of all subgenera of *Eucera* are present. It is nonetheless an assumption, and this could be discussed as a caveat related to the under-sampling. Moreover, I think the authors should run some basic simulations to at least suggest that this level of under-sampling should not bias their results. See some suggestions below.

3) Still related to the under-sampling, and the use of specific sampling fraction, I am not sure you need this analysis, certainly not as the main analysis. I say that because there has been some concern in the field that using specific sampling fraction might in fact bias the results and should not be done. Here is one such opinion: <https://cran.r-project.org/web/packages/hisse/vignettes/Clade-specific-sampling.pdf>. On the light of this potential problems, I would use the specific sampling strategy as a secondary analysis, and mostly rely on the global sampling strategy for the MCMC analysis.

4) Still related to under-sampling, I wonder on the effect of choosing equal proportions of traits for the missing species. If that is not the case, then the results could be quite different right? An under-sampling of only about 20% makes it difficult to consider if this is a reasonable assumption. Why choosing a sampling fraction of 50/50 percent? It might be seen as a conservative approach but if the frequency of R is in fact a lot smaller, then the results might be different. If I got it right, the “empirical frequencies” of those sampled species are: 47 only A, 11 that are A + R, and 22 only R. Hence if you consider “A+R” as “R”, your empirical sampling suggested that 33 out of 80 are R, about 41%. It might be interesting to run some sensitive analysis using slightly different frequencies. One could even test how skewed has the relative frequency has to be for the signal to disappear.

5) I think the authors need to run some specific simulations to show that their level of under-sampling/frequency of traits in missing species are not biasing the results. Additional to that some model adequacy simulations might be an interesting idea. Some ideas that came to mind (really not exhaustive and you need to think this through) are:

a) Simulate a tree of similar size to what would be expected if you have all species, using the estimated rates associated to each trait, remove species to represent the level of under-sampling, assume equal trait frequencies for the missing species, and then run BiSSE analysis to see if it can indeed recover those estimates. This is a “baseline” simulation that would prove the methods works well in this space of parameters. Not the most relevant to check potential biases but nonetheless a good start.

b) Run a simulation where there is no difference in diversification associated to traits, remove species to represent the level of under-sampling, assume equal rates of traits to the missing species, assume equal trait frequencies for the missing species, and then then run BiSSE analysis to see if it can indeed recover those estimates. This would test for some kind of “false-positive”.

c) I am not sure these are the best simulations, just wanted to exemplify some possibilities and encourage the authors to think about it. The following paper might give some ideas, and be a good start: Pennell, M. W., R. G. FitzJohn, W. K. Cornwell, and L. J. Harmon. 2015. Model Adequacy and the Macroevolution of Angiosperm Functional Traits. *The American Naturalist* 186:E33–E50.

6) The revised version now has only one figure, the phylogeny with the trait information. I agree this is a very important figure. I always like to have the data in some form, so the reader is able to gain not only information, but some intuition on what the results should be, in this case seen how the branching events are distributed through time and different clades. But after the revision the second figure showing the main result (difference in diversification rate) was taken away. Given that the main result is about diversification rate differences related to trait, I think a figure showing those differences (as it was the case in the previous submission) is necessary. I would therefore suggest the authors to present a figure where we can see the diversification differences.

Considering those aspects, I cannot recommend the publication of this paper in its current form. If the authors think they might be able to address these issues (including simulations to show how robust the results are to this level of under-sampling, as well as other methodological choices – e.g. equal trait frequencies for missing species), then a new submission might be considered. Otherwise, I suggest adding some caveats for the diversification analysis, perhaps change the focus of the paper away from the diversification analysis, and send it to a more specialized journal.

Reviewer(s)' Comments to Author:

Referee: 2

Comments to the Author(s).

Authors did a really good job in answering to all comments / recommendations from the referees and editor. I have one last recommendation but this is not critical (that's why I proposed a minor revision). As authors considered now the level of diet specialisation in their analyses of diversification rate (great!), I think that they should do the same for the analyses of Ancestral state reconstruction. They actually already discussed about potential results in the Discussion section (line 373).

Minor comments:

- Line 95: You should add a sentence to justify why you selected Eucera to explore this topic.
- Line 157: I did not understand how you recorded Fabaceae. You mean, all Fabaceae have restricted pollen type? Or you did not record any non-restricted Fabaceae pollen (like from Mimosa).
- Line 211: typo on performed
- Lines 269 and 275: I would say "36 species from early diverging lineages" and "44 species from recently diverging subgenera".
- Line 343: I did not understand the last part of the sentence.

Referee: 1

Comments to the Author(s).

I previously reviewed this manuscript (reviewer 1), and feel that the author's responses to my queries were largely satisfactory. I certainly defer to the editor's judgement for diversification analyses. I think the authors' inclusion of a few extra groups in the phylogeny (and not pollen data set) did little to address my concerns about completeness or ACR, but given that this is the data set that needs to be worked with – and is impressive – I do not see a real problem there. It is certainly easy to imagine that that the 80% of non-sampled species could shift, or even reverse, some of the conclusions. But that's how science works and is not a failing of the paper.

This revision, I delved into the plants, as that is where my expertise is most germane. The biggest issue is that the classifications are completely opaque – its not in the methods anywhere. This time I read through the supplement much more in depth – but not with a fine-toothed comb - and my biggest complaint is that you need to really describe how you came up with the restricted/accessible, oligolectic/polylectic classifications; they don't really align with what I would score from the data. There are a bunch where there were many known host families, AND you recovered multiple pollen, yet you classified them as oligolectic. Others where the fabaceous genera listed were not restricted.

I think you need to either assign uncertainty in those classifications (and bootstrap) or come up with some really strict guidelines that make sense. These will, absolutely, change the classification of some species, so the analyses will need to be rerun.

My other big issue is still lack of specific language.

2 - Title: I still disagree with the inclusion of flowers; there is no evidence that they diversify from bees in this paper. This title does not accurately represent the content of the paper.

31: In your response to my previous comments, you say that you changed pollinator to floral visitor throughout. You did not; it is used incorrectly here. (i.e. in 31, you really mean – insects which utilize floral resources – pollination has nothing to do with the diversification, your paper

is about floral resources). The preceding two are accurate, as you are focusing on the plant perspective.

34: Is it truly preference? Or just use/utilization? You don't have data on abundances to say they are preferentially using any species, just that they are using them (possibly exactly in line with abundance).

63: pollinators, again -> "those insects which utilize floral resources"

75: Why did they need to coevolve? I think you mean "evolved" here.

77: Bilaterally symmetric zygomorphic flowers need not have restricted pollen, nor does concurrent diversification imply coevolution. Change the sentence before and this will be more accurate.

97-98: "While all species can potentially extract nectar from bee-flowers" What does this mean? There are lots of generalized flowers that they can presumably get nectar from, too. "different sources of pollen" means specifically those plants that they do not nectar on? This section needs revision.

99-105: You are talking only about pollen collection here, right? Make that clearer before L 106.

105-106: "pollen specialists" belongs after "oligolectic species", as it defines it (same with generalists).

109: What trait? Specialization on a single family? Or accessing restricted pollen? Ambiguous as written.

111-112: "laborious work of palynological expertise". Nonsense as written. Do you mean "laborious work AND palynological expertise".

157: When you say only restricted pollen was recorded, do you mean you left out those species without restricted pollen when encountered or they never showed up in your dataset? They show up in your appendices as hosts, so you need to be clear here. (i.e. you have *Petalostemon*, *Dalea* as "restricted hosts" of *Xenoglossodes albata* - that's not correct at all - though *Psorothamnus* is - since you observed them on *Dalea*, how did you differentiate that 55% fab pollen into restricted? Plus the asters are all open).

257/258: Why pollen preference? Not pollen host or something else. Preference has a very distinct definition in foraging and that is not what you are using here.

269: Its not 36 early-diverging lineages - its 36 species from early diverging lineages, right?

272: *Zygophyllaceae* spelled wrong

275: there are not 44 early diverging subgenera.

288: "with high probability" belongs after inferred.

291-295: "preference to" -> "utilization of" (both the definition of preference, but also awkward phrasing).

331: not foraging behaviors - resource use.

332: delete comma

332-336: I don't understand the zygomorphy/actinomorphy part here. You are specifically talking about Fabaceae as the restricted/zygomorphic host, as borages are actinomorphic/restricted. I guess I think the symmetry is confounded with the fab/aster comparison (if Lamiaceae or Plantaginaceae were really big in the analyses, then sure, you could make a broader conclusion about symmetry, but they aren't).

338: morphological similarity of the bees or plants? Awkward phrasing.

346: is "higher Megachlidae" a normal phrase? I think evo. biologists (or at least botanists) are trying to get away from using higher/lower in favor of more specific/accurate phrasing (few say higher plants anymore).

375: They are "early-diverging" but since the species you sampled are modern, you can't really say they are basal.

385: pollen is known as a poor source of pollen? You mean nutrition?

388: Delete "Especially".

403: delete "unfavourable"

411: smaller than exactly what?

Figure 1: italicize specific names.

Supplement:

I didn't really want to go through this and I just skimmed it, but I guess as a plant person, I should note some that make no sense to me. They make no reasonable distinction between polylectic and oligolectic, there are botanical errors, etc. I think they should be changed or defended and possibly the analyses rerun. Honestly, you need to proof the list and maybe look at specimens of some genera to determine accessible/restricted. There are some genera that are considered restricted in one species and accessible in another (i.e. *Dalea* in *X. albata*/*X. eriocarpi*). Also some genera are assigned to the wrong family (again, *Dalea* in *X. albata*).

If your definition of oligolectic is a single family, as is stated in L. 105-106, the assignments don't make sense. Your actual criteria needs to be stated (see my broad comments above).

P. venusta/*P. carinata* – why is the former polylectic and the latter oligolectic? Both have the same a single family from pollen but more known host families.

P. carinata – lowercase specific epithet.

T. labrosa/nigripilosa – how is this oligolectic? 17% of pollen was from other families, and other families listed in the literature.

T. nigriceps – 40% of pollen is not from asters. This is not oligolectic.

T. tenuifasciata – again, how oligolectic?

X. eriocarpi – how is something with known hosts across 10 families oligolectic?

Xenoglossodes albata: This species is classified incorrectly as “restricted pollen” as most of the genera have their anthers out in the open, both the fabs (*Petalostemon*/*Dalea* – possibly referring the same species) and the asters (*Amorpha*, *Echinacea*) as “restricted hosts” of– that's not correct at all – though *Psoralea* is – since you observed them on *Dalea*, how did you differentiate that 55% fab pollen into restricted? Also, *Dalea* is listed as both an aster and a fab! It's a fab.

X. lippiae – again why oligolectic?

Synhalonia actiosa has *Zygophyllaceae* listed as pollen, but not as a host.

Synhalonia frater – *Hydrophyllum* is a borage, not a fab (and it is open, not restricted). There are no *Ericaceae* listed.

Synhalonia hamata – Some of the borages listed are open, as are the plantages (sure they have tubes, but the pollen is accessible). Also, with that host list, how is it considered oligolectic?!? You define that in the introduction as a single botanical family; even the pollen taken off is from three families. Sure, those two fell below the 5% threshold each, but together the other families were 9%.

Eucera dafnii – Neither restricted or oligolectic make sense with the entries. The only two genera listed are nonrestricted. Its not clear what the fab pollen came from, but I think you cannot say its oligolectic, even though the one sample had high fab pollen (also, not clear if restricted or not without a genus).

E. proxima – again, why oligolectic? *Anchusa* is accessible, I've no experience with *Hormuzakia*.

E. nigrilabris – again, why oligolectic? Looks like a lot of hosts.

E. clypeata – why oligolectic/restricted? *Caryophyllaceae* is almost certainly accessible (I think even many of tubed *Silene* have the anthers everted)

Author's Response to Decision Letter for (RSPB-2021-0533.R0)

See Appendix B.

RSPB-2021-0533.R1 (Revision)

Review form: Reviewer 1

Recommendation

Accept as is

Scientific importance: Is the manuscript an original and important contribution to its field?

Excellent

General interest: Is the paper of sufficient general interest?

Excellent

Quality of the paper: Is the overall quality of the paper suitable?

Good

Is the length of the paper justified?

Yes

Should the paper be seen by a specialist statistical reviewer?

No

Do you have any concerns about statistical analyses in this paper? If so, please specify them explicitly in your report.

No

It is a condition of publication that authors make their supporting data, code and materials available - either as supplementary material or hosted in an external repository. Please rate, if applicable, the supporting data on the following criteria.

Is it accessible?

Yes

Is it clear?

Yes

Is it adequate?

Yes

Do you have any ethical concerns with this paper?

No

Comments to the Author

I'm happy with the authors' responses and their reclassification of the botanical information; I trust that the authors went through all entries and not just the ones that I highlighted.

Review form: Reviewer 3

Recommendation

Accept with minor revision (please list in comments)

Scientific importance: Is the manuscript an original and important contribution to its field?

Excellent

General interest: Is the paper of sufficient general interest?

Excellent

Quality of the paper: Is the overall quality of the paper suitable?

Excellent

Is the length of the paper justified?

Yes

Should the paper be seen by a specialist statistical reviewer?

No

Do you have any concerns about statistical analyses in this paper? If so, please specify them explicitly in your report.

No

It is a condition of publication that authors make their supporting data, code and materials available - either as supplementary material or hosted in an external repository. Please rate, if applicable, the supporting data on the following criteria.

Is it accessible?

Yes

Is it clear?

Yes

Is it adequate?

Yes

Do you have any ethical concerns with this paper?

No

Comments to the Author

In this paper, the authors evaluate whether or not the preference for a specific type of flower changes diversification rates in eucerine bees. Despite the low sampling fraction, the authors run a series of tests to assess how their results are impacted by this lack of data, and show that their results are robust to many of these limitations. The study provides very important information not only to the knowledge of macroevolutionary patterns in insects, but also provides some insights on the role of mutualistic associations on the macroevolutionary dynamics of (at least one side of) the members of those interactions. Lastly, under a more methodological perspective, the supplementary analyses highlight that not all is lost with regards to the issues of trait-dependent diversification models, as long as researchers are careful enough to test the limitations of the results in each case.

I have very few minor comments that are described below, but the most important in my view would be to briefly address the expectations on how the prevalence between accessible and restricted pollen is in the major families of plants analysed in the study.

In general, I miss some further detail on the prevalence of bee-flowers/pollen accessible and pollen inaccessible flowers in the main families of plants, at least. This is important because even though switching to a new type of flower could represent an innovation, if there are only very few species with pollen inaccessible flowers the increase in resource might not be enough to spur diversification.

I commend the authors for the thorough exploration of the limitations in their results due to a rather small sampling fraction. It is always nice to see such care and honesty regarding the limitations in both the data and the methods, which in my opinion only make their paper more robust. The results are encouraging as to show that it would take a large asymmetry in proportions of states within the missing species to affect their results, but I don't know much about the particular system as to say if those values are unlikely or not (as the authors mention in the second paragraph of the "Assessment of the robustness of the diversification analyses to different assumptions regarding character-state distribution of the unsampled species" section on

page 3 of the SM). As mentioned in the previous comment, some further information on the expectations of this prevalence (such as a rough census of the prevalence of both types of pollen in the main families, for example) would solidify their results if they fall within the limits to which the methods look to behave well.

Lines 226-227: The posterior distribution in this case should be the distribution of differences, I didn't quite get the "percentage of MCMC samples"

Lines 326-337: So, in this case the authors are seeing differences in speciation rate, right? It is important to think about this, because if flower type is affecting differently only the speciation rates, there is some discussion that could be made regarding the possible mechanisms behind this. Also, in relation to this, the y-axis labels in figure 2 are misleading because it shows as them being the ratio instead of the difference in rates.

Lines 359-363: Couldn't the same driver be responsible for your results? It would be good to add whether or not Eucera-complex species include foreign material in nest construction as well.

Decision letter (RSPB-2021-0533.R1)

09-Aug-2021

Dear Dr Dorchin

I am pleased to inform you that your manuscript RSPB-2021-0533.R1 entitled "Bee flowers drive macroevolutionary diversification in long-horned bees" has been accepted for publication in Proceedings B.

The referee(s) have recommended publication, but also suggest some minor revisions to your manuscript. Therefore, I invite you to respond to the referee(s)' comments and revise your manuscript. Because the schedule for publication is very tight, it is a condition of publication that you submit the revised version of your manuscript within 7 days. If you do not think you will be able to meet this date please let us know.

- 1) A text file of the manuscript (doc, txt, rtf or tex), including the references, tables (including captions) and figure captions. Please remove any tracked changes from the text before submission. PDF files are not an accepted format for the "Main Document".

2) A separate electronic file of each figure (tiff, EPS or print-quality PDF preferred). The format should be produced directly from original creation package, or original software format. PowerPoint files are not accepted.

3) Electronic supplementary material: this should be contained in a separate file and where possible, all ESM should be combined into a single file. All supplementary materials accompanying an accepted article will be treated as in their final form. They will be published alongside the paper on the journal website and posted on the online figshare repository. Files on figshare will be made available approximately one week before the accompanying article so that the supplementary material can be attributed a unique DOI.

Sincerely,
Dr Locke Rowe
Editor, Proceedings B
<mailto:proceedingsb@royalsociety.org>

Associate Editor:

Board Member: 1

Comments to Author:

I commend the authors for all their hard work to improve the paper. The paper is now basically ready for publication with only some minor revisions as suggested by one of the reviewers. All of those are only wording and simple additions in the text to further explain a with things.

Reviewer(s)' Comments to Author:

Referee: 1

Comments to the Author(s)

I'm happy with the authors' responses and their reclassification of the botanical information; I trust that the authors went through all entries and not just the ones that I highlighted.

Referee: 3

Comments to the Author(s)

In this paper, the authors evaluate whether or not the preference for a specific type of flower changes diversification rates in eucerine bees. Despite the low sampling fraction, the authors run a series of tests to assess how their results are impacted by this lack of data, and show that their results are robust to many of this limitations. The study provides very important information not only to the knowledge of macroevolutionary patterns in insects, but also provides some insights on the role of mutualistic associations on the macroevolutionary dynamics of (at least one side of) the members of those interactions. Lastly, under a more methodological perspective, the supplementary analyses highlight that not all is lost with regards to the issues of trait-dependent diversification models, as long as researchers are careful enough to test the limitations of the results in each case.

I have very few minor comments that are described below, but the most important in my view would be to briefly address the expectations on how the prevalence between accessible and restricted pollen is in the major families of plants analysed in the study.

In general, I miss some further detail on the prevalence of bee-flowers/pollen accessible and pollen inaccessible flowers in the main families of plants, at least. This is important because even though switching to a new type of flower could represent an innovation, if there are only very few species with pollen inaccessible flowers the increase in resource might not be enough to spur diversification.

I commend the authors for the thorough exploration of the limitations in their results due to a rather small sampling fraction. It is always nice to see such care and honesty regarding the limitations in both the data and the methods, which in my opinion only make their paper more robust. The results are encouraging as to show that it would take a large asymmetry in proportions of states within the missing species to affect their results, but I don't know much about the particular system as to say if those values are unlikely or not (as the authors mention in the second paragraph of the "Assessment of the robustness of the diversification analyses to different assumptions regarding character-state distribution of the unsampled species" section on page 3 of the SM). As mentioned in the previous comment, some further information on the expectations of this prevalence (such as a rough census of the prevalence of both types of pollen in the main families, for example) would solidify their results if they fall within the limits to which the methods look to behave well.

Lines 226-227: The posterior distribution in this case should be the distribution of differences, I didn't quite get the "percentage of MCMC samples"

Lines 326-337: So, in this case the authors are seeing differences in speciation rate, right? It is important to think about this, because if flower type is affecting differently only the speciation

rates, there is some discussion that could be made regarding the possible mechanisms behind this. Also, in relation to this, the y-axis labels in figure 2 are misleading because it shows as them being the ratio instead of the difference in rates.

Lines 359-363: Couldn't the same driver be responsible for your results? It would be good to add whether or not Eucera-complex species include foreign material in nest construction as well.

Author's Response to Decision Letter for (RSPB-2021-0533.R1)

See Appendix C.

Decision letter (RSPB-2021-0533.R2)

27-Aug-2021

Dear Dr Dorchin

I am pleased to inform you that your manuscript entitled "Bee flowers drive macroevolutionary diversification in long-horned bees" has been accepted for publication in Proceedings B.

Data Accessibility section

Open Access

Paper charges

Sincerely,
Editor, Proceedings B
<mailto:proceedingsb@royalsociety.org>

Appendix A

Associate Editor

Board Member: 1

Comments to Author:

In this paper the authors investigate a biologically very interesting question, whether the interaction between plants and pollinators might have shaped the pollinator's macroevolutionary dynamics, in this specific case the bee genus *Eucera* (Apidae). It collects an impressive data set, uses the most recent phylogeny and do take into account several methodological aspects. That is why I was initially very excited about the paper, but after reading the paper a second time, I think there are some important methodological aspects (see my comments below) that could impar the most relevant result. Reviewers raised some really relevant aspects that could be addressed but the methodological aspect I will briefly explain will require a reanalysis of the data and no guarantee that the results will remain the same. My gut feeling is that the results will change (no evidence for differential diversification rate, or at least not able to reject the hidden state), but I could be wrong.

We thank the editor for the overall positive feedback on our manuscript and for the insightful suggestions.

My main concerns relate to some the "side effects" of having a smaller sampled size (about 20% of the bee species are sampled) when doing diversification analysis. Sometimes (most time) this is the reality of the data, and in fact the authors have gathered an impressive dataset on pollen, so I have to admit that I am thorned to be criticizing this aspect, but as I will explain I suspect that the results might have emerged from those "side effects" and some methodological choices. I commend the authors for taking several steps to mitigate and explicitly deal with these "side effects" (e.g. incorporate the sampling fraction on BiSSE and HiSSE) but on a second thought I think there are still some relevant aspects. I will try to explain my reasoning here and I hope this helps the authors in future submissions of this manuscript.

1) In the methods section the authors say they used the diversitree function "make.bisse.uneven" to inform the analysis what is the percentage of missing species associated to each subgenera. If I get this right, the BiSSE analysis attributes this percentage of missing species to the crown age of the clade of interest. The problem with this is that under-sampling is likely to affect the age estimate of the crown group itself, and hence the posterior estimates of rate. This is particularly likely when the under-sampling is high, which is the case for several subgenera here. Hence differences in rates might be strongly related to differences in how far each lineage is from the "true" crown group age and how well sampled is each subgenera. This is ameliorated by the very clever idea to use the same frequency of equal states for the missing species within each genus, but it is still possible that an effect is relevant here.

We agree that our study would benefit from higher clade representation but as discussed below, we believe that adding more species will not make significant changes to the tree topology and ages of early nodes in our phylogeny. First, we note that the uneven sampling option of BiSSE do not place the missing species under the crown node of the clade of interest, but rather assumes that diversification happened at any point after splitting from its sister clade, thus accounting for the possibility that the inclusion of the missing species could change the age estimate of the crown group itself (page 598 in FitzJohn et al. 2009, Estimating Trait-Dependent Speciation and Extinction Rates from Incompletely Resolved Phylogenies. Syst.Biol. 58. doi:10.1093/sysbio/syp067). Second, we note that the dataset adopted from Dorchin et al. 2018 was designed in the first place to represent as many different taxa as possible, namely *Eucera* genera, subgenera and morphologically divergent species

groups. We added a clarification in the 'taxon sample and dataset' of the methods section to specifically refer to this issue. In line 129 we write: "While this selection of 89 species sums up to 22.4% of the 397 *Eucera* complex species (based on Ascher and Pickering 2021), it adequately captures the taxon diversity of this clade."

In addition to the above, we have taken several measures to fully explore the concern raised by the editor. First, we have made efforts to add sequence data for as many eucerine species as possible via additional searches in Genbank and in BOLD (Barcode Of Life Data System). This search has yielded (high quality) sequences for additional 41 species, not yet included in our phylogeny, all which were of the mitochondrial gene COI. Except for one (belonging to the species *Eucera floralia* (Smith)), they were relatively short sequences up to about 600 base pairs in length, representing the so called 'barcoding sequence'. We then assembled a new dataset adding these sequences to our 'backbone' phylogeny of nucleotide sequence dataset, and performed a series of exploratory ML phylogenetic analyses. Accordingly, we found that the sequences 'correctly' placed (most sequences) were clustered within subtrees already present in our phylogeny and thus their inclusion is not expected to change the corresponding crown age estimates. However, as expected from their short lengths, the addition of these sequences resulted in considerably lower support values for the older nodes (bs support values below 60 and often much lower), as was found in other studies (for example see in Trunz et al. 2016, Comprehensive phylogeny, biogeography and new classification of the diverse bee tribe Megachilini: Can we use DNA barcodes in phylogenies of large genera? Mol. Phyl. & Evol. 103, <http://dx.doi.org/10.1016/j.ympev.2016.07.004>). This would not allow to calculate clade ages and estimate clade diversification rates, and therefore not useful to us beyond this confirmational analysis.

Second, compared to the original submission, we have succeeded to expand the phylogeny by 17.1%. Specifically, we were able to add 8 *Eucera* species that were included in the phylogeny of Dorchin et al. 2018, for which we have full sequence data (but no pollen data), plus *E. floralia* mentioned above. We further expanded our dataset by adding four eucerine species in the early diversifying genera *Simanthedon* and *Protohalonia*, representing 100% of the species in the sister group of the genus *Eucera*, for which we obtained pollen data from published studies and from new pollen analyses we performed. Our expanded dataset includes altogether 89 species, of which 80 species have pollen data, and that represent 12.7-100% of the species in each of the clades. The new data assembly is now described under the "Taxon sample and dataset" section.

2) More importantly, when doing the HiSSE analysis, which is extremely relevant to discard the potential effect of a hidden state driving the differences detected on the BiSSE analysis, the sampling fraction of missing species used was not for each subgenera (as for the BiSSE), but for the "whole phylogeny", which is not a problem per se (in fact I think this is the only automatic option for HiSSE), but there is one potential problem in the context of the paper. Given that about 80% of species are missing, it is possible that the choice of the BiSSE model over the HiSSE model was somehow affected by differences in how the sampling fraction was implemented on the phylogeny. This is because in the hidden state model most of the potential rate heterogeneity was inserted as "white noise" in a sense (80% of species added with equal frequencies for each state throughout the whole tree). Hence it is possible that a better fit for the BiSSE model (which is the key result to suggest that the variable of interest, the type of pollination, affects diversification rates) results from this difference on how to handle the missing data. This effect might be exacerbated by the potential effect mentioned above on point "1".

SUGGESTION: A better comparison would be to compare the HiSSE model to the BiSSE model where the sampling fraction is not informed clade by clade but for the whole phylogeny in both analyses. In this

way the comparison is more adequate and the potential problem associated with the inclusion of species in the crown age disappears. My intuition is that this will remove the statistical difference between HiSSE and BiSSE, but I could be wrong. Alternatively, if the authors devise a way to include the missing species through the whole stem lineage (not only through the crown group) and do that for both the HiSSE and BiSSE analyses, I think this might be another way to more appropriately compare the results and rule out the effect of a hidden state.

We believe this concern stems from our failure to describe the exact procedures that we have previously employed, which were essentially very similar to the suggestion of the editor. Specifically, we have conducted two main analyses: one based on an ML approach and another using a Bayesian MCMC procedure. These two analyses were conducted using two different R packages, thus taking advantage of the unique advanced options available in each package. First, the entire ML analysis was conducted under the same package. This package allows the examination of models that incorporate hidden states. However, this package supports only global sampling and not clade-specific sampling. Accordingly, in all ML analyses we have not partitioned the data by clades but applied a uniform global sampling frequency across all terminals and character states. Thus, in all models examined, with and without hidden traits, data were partitioned in the same way. Otherwise, just as the editor had noted, the likelihood of the different models could not be correctly compared.

In an effort to further improve our analysis, we have now performed the ML analyses with the recently developed SecSSE package, which presents somewhat improved capabilities compared to the HiSSE package (Herrera-Alsina et al. 2018, Detecting the Dependence of Diversification on Multiple Traits from Phylogenetic Trees and Trait Data. *Syst Biol.* 68(2): DOI:10.1093/sysbio/syy057. In this method, models equivalent to BiSSE and HiSSE are implemented, but unlike the HiSSE framework, this method allows for estimating both the speciation and extinction rates, and thus to examine models in which only the speciation rates are free to vary while the extinction rates are constrained to be equal, such as we have done. (In the HiSSE framework these rates are combined as a new parameter - the net turnover and extinction fraction.) Similarly to HiSSE, all models in SecSSE include hidden traits and the program also allows for applying a global sampling fraction parameter.

We have also repeated the Bayesian MCMC BiSSE analyses, and compare analyses with global sampling fraction as in the ML analyses to analyses with clade specific sampling fractions as we have done before. We find the results from these two sets of analyses are nearly identical, and are largely in line with those of ML analyses. Specifically, our new analyses show that the Full state-dependent BiSSE model and a BiSSE model with extinction rates constrained to be equal have the highest support and both suggest significantly higher net diversification rates of eucerine bees under the 'restricted' state. In these BiSSE analyses we partitioned the data and applied a sampling fraction parameter for each clade.

Additional comments/suggestions are:

A) A bit more information about the lineage (could be very short) before the discussion would help the reader visualize the study/sampling methods (this was also mentioned by one reviewer). For example, where is the genus geographically found? Is the pollen data derived from specimens collected through the whole genus distribution or for only parts of it? Some of that information might be in the supplemental material but a few sentences in the main text might be helpful.

Done. We have added the relevant information in the Methods section 'Determination of pollen types'. We now emphasize that our pollen analyses include samples from all major distribution regions of the studied eucerine clade. We also refer the readers to Table S1, where the distribution of each of the species sampled is given

together with the number of samples and the number of discrete sites. Because our main purpose is to make the distinction between accessible and restricted pollen types, and not necessarily to accurately determine the pollen diet breadth of each species, we would prefer not to add long locality descriptions in table 1 that already contains a lot of data.

B) Would it be interesting to try a fourth character data set where species classified as “Accessible + restricted” were considered “Accessible”, and the species classified as “restricted” was scored as R in the data matrix only if had restricted? In the first character data set the authors scores A if species has only A, and R if species has A + R right? I understand the reasoning here (which could be made explicit in the main text), but I wonder if a fourth scenario (doing the opposite scoring scheme) would not help reinforce that the R state is what is driving the increase in diversification rate. My concern is that the higher rate of “R” might be influenced by the rates experienced by those species that have both R + A. Here it would behave as a hidden state. I know the HiSSE model was designed for that but given that under-sampling was treated differently (see comment above), this would be another potentially interesting test. Doing this alternative scoring might not be that relevant (in particular in the light of the second score matrix where some species are scored as ambiguous), but an explicit explanation why R + A was coded as R in the first data matrix might help the reader.

Done. Following the suggestion of the editor, we have added a character set with the ‘polymorphic’ species treated as accessible. Both ML and MCMC analyses showed the ‘restricted’ state had higher diversification rate than the ‘accessible’ plus the polymorphic states, giving support to the hypothesis that association with restricted pollen, but not with the combination of restricted+accessible pollen, is a factor driving increased diversification rate in eucerine bees. We also added a sentence to explain the reason for categorizing R+A pollen as R, and nevertheless mention the additional analyses performed with different categorizations of the polymorphic state. In line 236 we write: “this categorization of the polymorphic state is the most reasonable one, since species that are scored as R have the ability to extract restricted pollen, but we nevertheless examined the following alternatives”.

C) This is a minor fix, but I was initially a bit confused on how the MCMC result on diversification rate was evaluated. When I got to figure 2 I understood it better but a better explanation might be necessary at the methods. For example, it is said (lines 206-208) that “The posterior probability that state R exhibits higher net diversification rate than state A is represented by the percentage of MCMC steps in which $dR > dA$.” This gave me the initial impression that is the frequency of times the MCMC finds a higher values for dR rather than the magnitude of the difference in $dR-dA$ (shown in figure 3) that was used as evidence in favor of $dR > dA$. I guess both aspects were used (frequency and magnitude) but somehow reading the text I felt that the magnitude aspect was not explicitly presented. Just make it more explicit in the methods section. E.g. you will plot the posterior probability of differences as shown on figure 2

Indeed, the main statistic that we evaluated was the fraction of MCMC steps in which $dR > dA$. Following this comment, we now also provide more information regarding the magnitude of the difference in net diversification rates between the restricted and accessible state. Specifically, in line 312 we provide the information regarding the average difference ($dR-dA$) and the average ratio (dR/dA) across all MCMC steps. Results of both statistics are also illustrated and compared in figure S1 in the SM.

Given the issues raised above, and those raised by the reviewers, I unfortunately cannot recommend the paper for publication in its current form. If my impressions regarding the differential treatment of missing species (BiSSE vs HiSSE) is correct, it will require a reanalysis of the data which might substantially change the results. Do not get me wrong, I think the work is interesting, the data set amazing, and it

should be published, but depending on the new results, perhaps in a more specialized journal and/or focusing on other aspects rather than the diversification rates. If the authors think they can properly address those issues and the main result are still interesting to a broad audience, then a resubmission would be welcomed. Given the uncertainty of the re-analysis and that the results (or focus of the paper) might substantially change, I think this potential resubmission should be treated as a new submission.

We thank the editor for the many insightful suggestions. We believe we have considered all concerns raised by the editor and the reviewers. These have helped to improve the analyses and to clarify the text, particularly with regard to the way the ML analysis was performed.

Reviewer(s)' Comments to Author:

Referee: 1

Comments to the Author(s)

Review of Dorchin et al.

This study is great effort to get at a question which ought to be interesting to a great deal of researchers. As a person strongly on the plant-side of pollination, I was unaware that there was little work on diversification of bees in response to floral morphology. The pollen data which they assembled was surely a great deal of work and the choice of a group of bees and the sampling (20% of the clade) is quite high for these larger-scale ecological comparative analysis. I think this paper is certainly of the suitable for publication in Proc B after revisions and I am sure it will be of interest to many.

We thank the reviewer for the positive feedback on our study and for the appreciation on the hard work conducted. We also thank the reviewer for the many insightful suggestions that further helped us to improve the manuscript.

I have pages of individual comments, but they broadly fall into:

1) Clarity issues – there are a lot of methods and results which are not explained sufficiently to understand exactly why you chose to do them. For instance, why is pollen volume used and how is it calculated from the percentages gathered? And if the bee was collected early in a foraging bout, why should a lower volume of pollen be any less indicative of preference?

Some authors decided to consider scopal pollen load in quantitative pollen analyses, such as in Sedivy et al. 2013, which we have used as reference, others have decided not to use it. To our understanding, the idea behind weighting pollen samples by loads is that even among generalist (polylectic) bee species, individual females tend to collect pollen from the same flowers on the same foraging bout (sometimes referred to as 'floral constancy'), such that small pollen loads will be more homogenous than larger loads, and prone to represent less well the pollen spectrum used by that species. Given the page restrictions, we prefer not to describe these methods in detail in the main text and refer the readers to a more complete description provided in the SM.

2) There are a few long paragraphs with multiple ideas that need to be shortened and clarified (i.e. 376-417).

3) Many references are needed throughout for various statements and in some cases, references should be checked for accuracy (i.e. final paragraph).

4) Certain word choice is vague and needs to be made much clearer: e.g. "Evolutionary success" is used to indicate an increase in diversification rates.

5) The final paragraph about the evolution of the specialist group and the flowers they rely on needs to

be significantly rethought and re-referenced.

Please see out reply below to the more detailed comments.

Title: It seems like the novelty of this study is that you focus on the bees, yet the title says “bee flowers” first, and until I read the abstract, I thought this would be a plant paper. I’d rephrase somehow to stress the bee aspect. (Even, perhaps, just reversing bees and bee flowers).

Done - modified as suggested. We still kept ‘bee-flowers’ in the title to put some emphasis on the interaction.

30-31: I’d be a little more circumspect here (and elsewhere) with word choice – you don’t really know that these bees (or any floral visitor!) is truly a pollinator, or if a pollinator, a good one or one important to the plant. You are very careful with this elsewhere (i.e. the nice system/study description in L95-115). It is worth being very specific, even in these broad sections like L30-34, as the bees needn’t be pollinators to have diversifying in response to floral preference/floral traits.

“Floral visitor” is the phrase I use most often, but that’s not really appropriate here – perhaps “pollen consumer” or something similar. Loads of even specialist bees/moths/etc, are not particularly good pollinators, but are reliant on one or few plants.

Agreed, we have used ‘floral visitors’ throughout the text.

55-56: The clause “and many floral traits are used in the characterization of angiosperm species” is a non sequitur here, and interrupts the flow between pollination ideas. I’d delete or rephrase this sentence.

Done. We deleted this sentence.

57: What exactly is the assumption here? That pollinators shape floral evolution? I think deleting the clause will make this clearer, but it might be worth rephrasing this section.

Done. We modified the section as suggested. In line 54 we now write: “It is generally accepted that pollinator mediated selection has been central in shaping floral evolution (Johnson 2010).”

56-65: You definitely don’t have to take this suggestion, but in the two sentences before the concluding one, you discuss a lot of pollinator specialization (which implies some evolutionary knowledge), then say that there is little known about the evolution of pollinators. As I understand the literature, the diversification of plants after developing things like nectar spurs or long tubes is not actually demonstrated from pollinator specialization (i.e. most hummingbirds and hawkmoths are not specialists, in the sense that they hit a single species or genus of flowers), instead from these plant morphological changes decreasing the generality of their flowers; i.e. the plant is specializing, not really the pollinator (which you get at in L58-59), but then you switch to pollinator specialization and repro isolation in the next, and conclude that little is known about evolution of pollinators.

We thank the reviewer for this comment. In this paragraph it is meant that the pollinators became associated with specialized flowers, such as with floral tubes and spurs, rather than visit any flowers. Thereby they contributed to floral isolation and ultimately speciation. We modified the sentence to make this clear. In line 58 we now write: “Increased specificity of floral visitors that are associated with these specialised flowers has promoted speciation of plant lineages via floral isolation, although other isolating factors also contributed to their speciation (Crepet and Niklas 2009, Kay and Sargent 2009).”

76-78: Might be nice to have a reference here: bees could certainly exploit flowers for both pollen and nectar without being long-tongued or co-evolving with the flowers (lots of native bees here nectar and

collect pollen of non-native flowers, including restricted zygomorphic species, with which they have no co-evolutionary history at all).

Done. This particular sentence refers to the historical evolutionary relationships between bees and flowers, rather than the modern associations we see today. We make the assumption that the long tongue bees developed the ability of extracting restricted nectar as well as pollen from morphologically specialized 'bee-flowers', and that this has led to coevolution between these groups. This does not mean that some modern short tongue bees have not developed adaptations to exploit the same flowers for pollen as well, or that some modern long tongue bees do not collect floral rewards from other flowers. We modified the text to explicitly mention this in the same paragraph. We provide a reference to support our assumption in the subsequent sentence.

81-83: This sentence about exceptions in bees, needs context, since it follows a plant sentence. What exact exceptions do you mean? What flowers they exploit? The bee morphology (i.e. non-LT bees exploiting zygomorphic flowers)?

Done. We modified this paragraph. In lines 79 we now write: "Yet, diverse adaptations to extract restricted pollen are known among modern bees, including also short tongue bees (Michener 2007: chapter 19, Danforth et al. 2019, chapter 7)."

94: "potential evolutionary advantage" What do you mean by this statement? Later on, in your results, you use this to mean increased diversification, but this is distinctly not what you mean here. Rephrase or define exactly.

Done. This is now clarified in the text. In line 90 we now write: "Further evolutionary innovations, namely flower buzzing (Cardinal et al. 2018) and utilisation of floral oils (Aguiar et al. 2019), were not found to significantly increase bee diversification as well, although they contributed to increase species richness in the former and habitat occupancy in the latter."

95-115: This is a great overview, but can you detail a little bit of their interaction (i.e. they collect pollen, do they also drink nectar? You collect pollen from the scopa later, but you mention nototriby in the intro – do they also get pollen in other parts of the body or do they groom it down right away?). This is probably all known by most bee people, but this paper should attract attention of a lot of plant people, too, so a little more natural history of the bees would be nice for those of us who don't know bees as well.

Done. We added to the paragraph the most important information for understanding of the study system, but unfortunately we could not expand on the specific pollen collecting behaviors due to page restrictions. The interested readers may find this information in the provided reference (Michener 2007: chapter 19, Danforth et al. 2019, chapter 7, Portman et al. 2019, Buchmann 1983, Westerkamp 1996, 1997, Müller et al. 1996a, 2006, Westerkamp and Classen-Bockhoff 2007, etc.)

102: Boraginaceae are actinomorphic and often shallow. I know that some are deep and specialized, but many readers will probably picture something like Heliotropus or Nama, which are actinomorphic and probably easily exploited by many floral visitors.

We thank the reviewer for pointing us to this point. We have modified the text to "some Boraginaceae" (line 104), and mention that we partitioned between Boraginaceae pollen types of contrasting accessibility in determination of pollen type of the methods section. We do not list in the main text the different restricted and accessible pollen types of Boraginaceae that we recorded, but these are presented in detail, together with our classification to

accessible or restricted pollen in table S1, and we refer the readers to the table in the text.

119: How many species total are there in *Eucera*? For the 85 species, how much of the radiation do they cover? This is at first glance a good sample size for comparative analyses (especially that you sampled pollen from 76 of the 85 species!!!), but it is worth taking the cautions detailed in Losos (*American Naturalist*, 2011, 177: 709-727) into consideration: i.e. if those 85 are drawn from across a phylogeny of 1500 species, there are some caveats to that analysis (in L95-96, you just write “one of the largest”, and as a plant person, I have no idea if this is 100, 500, or 1000 and that surely matters to interpretation). Later on you clarify that it is 20%, but this needs to be noted here, too.

Done. We have added this information to the methods. Given the comments of the editor and reviewers, we have tried to increase our sample size by adding 9 *Eucera* species for which we have full sequence data but no pollen data (therefore not included in the original analyses). We also included additional four species that comprise the sister group of *Eucera*, collectively referred to the '*Eucera* complex' (or the 'eucerine clade'). Altogether our phylogeny includes 89 species out of 397 species known in the group or 22.4% (20.1% if only the species with pollen are counted).

131-134: What a dataset!

Thanks!

139: Why are percentages per species summed? I do not think you can sum percentages (i.e. 50% + 75% is uninterpretable); did you mean that they were they averaged? If that is what you mean, is this appropriate without some sort of weighting (i.e. if you had 1000 pollen grains on one bee, should that be averaged with another bee of the species which had 10 pollen grains?)

We agree with the reviewer that the terminology that we previously used was confusing and this was modified to better explain what was performed. Specifically, the percentages were not simply averaged but the percentage of each pollen type was corrected based on the size of the original pollen load (percentage multiplied by size of pollen load). This product should better be called 'weight' rather than average. So, we actually placed a weight on each pollen sample, summed all pollen weights together and divide by the total weight to give the final percentage for the species. We practically adopted the same method used in Sedivy et al. 2013 and a number of comparable works. Using the new terminology, this is now explained in the SM (under Quantitative methods of pollen analyses).

138-145 (and again 262-279): It seems as if you are calculating percentages of pollen from three transects across the slide. However, you then switch to pollen volume without giving a good idea of (a) how you calculate this (something about the scoring of pollen load size – also should note is full load or 1/5 load consistent in volume across species?) and (b) why this is a better metric than just % for pollen.

Same answer as above, pollen percentages were given weights, which were then summed across all samples and divided by the total weight. We explain the quantitative methods of pollen analysis in the SM, which should be clearer now. The estimation of pollen load used for placing weights on pollen samples was done qualitatively prior to pollen extraction and the rationale behind this is explained under point 1 above.

156: “both pollen types”? Did you mean “pollen from both of these plant genera”?

This referred to the Cucurbitaceae and *Ipomoea* pollen types. Following text modifications, in the revised manuscript this sentence was deleted.

216-217: Detail what “remaining taxa combined as background” means. From Figure 1, I can’t really tell what “background” means (I thought perhaps outgroups, but maybe you mean the small clades?).

Yes, this means the small clades were classified together for the purposes of the statistical analysis. We changed now to ‘terminals’ to make this clear (line 232).

249-250: I have done relatively little modelling of this sort, but I would be somewhat concerned if the models are estimating negligible extinction rates – do you mean fairly equal across clades? Or, as written, that there has been basically no extinction? Or something else entirely? Clarify.

Our ML and MCMC analyses show the extinction rates were much smaller compared to the speciation rates, leading to diversification rates that are comparable to the speciation rates but still not identical. Tables S6 shows the diversification rates are slightly lower with the first four datasets. The low extinction rate may be due to the fact that the eucerine clade is relatively young (23 My), especially the genus *Eucera* that comprise the majority of the species (16.6 My). That said, it is widely accepted in the community that extinction rate estimates should be interpreted cautiously since these are particularly sensitive to sampling biases and departures from the assumed model (e.g., Rabosky 2010, Rabosky DL, 2010. Extinction rates should not be estimated from molecular phylogenies. *Evolution* 64. doi:10.1111/j.1558-5646.2009.00926.x). Thus, the primary focus of our analysis was to compare the net-diversification rates of bee lineages relying on restricted versus accessible pollen, rather than to tease apart the relative contribution of speciation and extinction. Accordingly, as we do not feel that current methods can robustly tackle this point, we prefer not to discuss in the text whether the increased net diversification rates of R lineages is due to their higher speciation rates or to lower extinction rates compared to A lineages.

262-279: Pollen volume? From the methods, I thought you counted grains across three transects on a slide and used percentages of the total. Detail better in methods the conversion to volume.

Done. Because we counted pollen grains and not calculated their volume, we agree that this term cannot be used and changed ‘pollen volume’ to ‘pollen content’ throughout the text.

344-345: Can you describe what instead they thought drove this pattern? Or was it simply untested? Either way, it should be noted.

Done. We added this information to the text. In line 346 we write: “They however considered another behavioural innovation as the main driver of diversification, the inclusion of foreign material in nest construction.”

347: “and in fact they are more frequent than previously thought”. Either a reference to say that these shifts are rare, or just delete the clause. Not sure it adds that much – you convincingly show shifts and I trust that the other references you cite do, as well.

Done. We deleted this clause as suggested.

351: I would avoid “Specialised” here and change it to “inaccessible” as you have elsewhere.

Specialised, to me, means relying on a single or few pollinators, which is not what you mean here, I believe. Nobody argues that *Trifolium* is specialized – though it is bee-pollinated, for sure!

Fixed.

352: I think you have the tribes wrong, and your references do not support your assertions. I thought *Trifolium* was *Trifolae*? *Astragalus Galegae*? I know these are all closely related and I don’t have access

to the Lewis book you cite, but the Sanderson and Wojciechowski paper has a bewildering array of tribes in there. Anazi (2019) the most recent reference that you cite does not make any mention of tribes, but cites Schaefer et al (2012), which says that Fabaeae is just Pisum, Lens, Vicia, Lathyrus and Vavilovia. If they are monophyletic, note that, but I don't think these are all Fabaeae, and therefore this tribe certainly does not represent "the great majority of temperate herbaceous Fabaceae".

Fixed. Indeed, the Fabaeae is only one of the tribes included in the so called inverted repeats lacking clade (IRLC), which forms a monophyletic group in Lewis et al. 2005 as well as in Sanderson and Wojciechowski 1996, although less clearly delimited in the latter. We meant to the IRLC and actually first used this name and erroneously changed to Fabaeae at some point.

363: "were" to "where". It is worth noting that Fabaceae are abundant and widespread everywhere, and most diverse in the tropics, so more explaining might be helpful – i.e. Trifolium/other herbaceous fabs might be what you are specifically discussing?

See our reply above. Changing to IRLC should make this clearer.

372: Might be nice to note that you, too, recovered that association.

Dorchin et al. (2008) inferred the ages of diversification and geographic origin of the different eucerine clades, and in this study we reconstruct their historical association with floral hosts, which was not known previously. We have modified the paragraph to make this clear. Specifically, we write in line 358: "*Eucera* and *Synhalonia* diversified in temperate regions of the Holarctic (Dorchin et al. 2018), where the IRLC taxa are abundant and widespread, and we show they developed association with many of these plants as preferred or exclusive host plants."

372-373: Why is the *Melissodes* preference important here? And how does it lead into the concluding sentence? *Xenoglossodes* is nestled in your clade (even if paraphyletic), *Melissodes* would be a distant outgroup and would say nothing more about *Xenoglossodes* than any of the other groups. I'd either explain the link you are trying to make here more clearly or delete.

373-376: Sort of the same as the previous comment, but why does radiation on Asteraceae for *Melissodes* (and the diversification of *Tetralonia* and *Xenoglossodes* not on Fabaeae) set up the *Synhalonia* radiating with the switch to restricted pollen? Make this abundantly clear.

In the resubmitted version this part was deleted due to space constraints.

381-402: Your ACRs (presented in Figure 1 and tables S3) are about accessibility, not on plant family. Looking at Figure 1, there are many asters represented, but also other families of accessible pollen (i.e. cucurbits, malvs, mustards, etc.). You cannot say that they have "aster specialist ancestors" or that Asteraceae is the ancestral host (i.e. it looks like *Tetralonia* probably has aster as an ancestral state, but it seems unlikely that the other clades do).

Fixed. We deleted "ancestral". We previously used this term to emphasize the high frequency of Asteraceae pollen types among the early diverging lineages but we agree that this term is incorrect.

412-417: This conclusion is a non-sequitur from the pollen chemistry focus of the paragraph. I think I might put it above the pollen chemistry (The flow would then be: Unobserved hidden traits had an effect.... One very likely factor would be pollen chemistry).

414-415: This point needs to be detailed a bit better, with references and some interpretation. This is why I asked for the sampling and total species numbers in the description of the system, since a lower

sampling amount (and 20% is actually pretty good!!!) is a caveat, and not only to SSE models, but to all the interpretation – even ancestral states can be strongly influenced if sampling is low (and there may be this issue for the *Eucera* section, as you note it evolved in american deserts, yet your sampling is entirely palearctic for this group).

Done. We have added the required information, and explain why our dataset can still be used for the purposes of this study in the 'Taxon sample and dataset' of the methods section. We thus deleted this sentence because it largely repeats the same message. We also added description of the geographical distribution in the 'Determination of pollen type' of the Methods section. The name *Eucera* is particularly confusing, because the genus *Eucera* (as given in Dorchin et al. 2018) is inferred to have originated in the Nearctic region, but the subgenus *Eucera* is endemic to the Palaearctic region, and this is the reason that all the species sampled are from that region. This should be a bit less confusing because we now refer to the entire clade as the *Eucera* complex or eucerine clade, following the inclusion of two sister lineages of the genus *Eucera* to our dataset. These lineages are both also endemic to the Nearctic region.

374/418/422: “advantage”/”advantageous” rephrase and be specific, i.e. higher diversification rates – I don't think that a higher diversification rate is necessarily an advantage in an ecological sense (and seems nonsensical in an evolutionary sense with lots of switches).

Done. We have modified the discussion section and deleted these terms.

424: delete “speciliases or” – it reads more clearly as “bee fauna largely depends on a few...”

Done.

426-430: This whole paragraph needs to be rewritten and rethought, as these references do not support your statements at all. The 2000 Minckley paper discusses specialization on *Larrea* – an actinomorphic, open flower – and the 2013 paper discusses *Larrea* and *Prosopis* (an accessible, with pollen in the open, woody, fab). You do not mention exploitation of *Zygophyllaceae* at all and you make it seem like all *Fabaceae* hosts are herbaceous and with restricted pollen. Herbaceous species in deserts are not reliable and abundant (though they may be the latter in certain years), and your assertion that “association with these host plants were maintained because they offered abundant and reliable resources” is not at all supported, since the abundant and reliable host genera (i.e. those dominant desert shrubs) are not in your host list at all (creosote shows up once in your reported host list; mesquite not at all).

Looking through the SI material, the one apparent desert species (e.g. *Synhalonia primiveris*) are primarily feeding on the exact opposite of this assertion – unreliable annuals (*S. primiveris* does feed from *Psoralea*, *Hyptis*, *Prunus*, *Purshia*, *Larrea* and *Tamarix*, but those are little of the pollen you recovered). Despite your assertion that *Eucera* species “evolved in the xeric habitats of North America” (L423) you sampled only Palearctic species. (I am assuming that you mean the *Eucera* clade, as indicated in 419, not the whole genus)

428-431: The abundant and reliable species are not the restricted ones, as implied.

436: Again, be more specific instead of “evolutionary success” – increased diversification rates.

After adding other content to the discussion, we decided to delete this paragraph, which is less relevant to the rest of the Discussion. We previously used the Minckley et al. works to demonstrate that regardless of pollen accessibility mode, the bees will retain their floral hosts as long as they remain abundant and reliable pollen sources. This not necessarily relate to the actual pollen types that we recorded in our study.

Regarding the Fabaceae host plants in our dataset, all of the pollen types we recorded were indeed representing restricted pollen hosts. We explicitly mention this now in 'determination of pollen type' of the methods section. In line 156 we write: "In Boraginaceae, different pollen types were partitioned according to floral accessibility, whereas in Fabaceae, only restricted pollen types were recorded and were pooled together."

Figure 2: I might put a line at 0, just to indicate where the no effect would be (and that your most of the probability distribution under both models is far above that point). I think that would make your result a bit more striking.

We thank the reviewer for this suggestion, and in the revised version of our manuscript present the new figure as a supplementary figure S1 in the SM, with lines added in the plots as suggested.

Referee: 2

Comments to the Author(s)

Authors present a very interesting study on the evolution of a species diverse group of solitary bees (i.e. genus *Eucera*) in relation to their pollen diet. Overall the manuscript is well written, simple and easy to follow (but see some comments below). The references are accurate, the analyses are well done and the discussion is well balanced. This article will interest a broad audience of evolutionary biologists. I recommend its publication.

We thank the reviewer for the positive feedback and for the many insightful suggestions that helped us to further improve the manuscript.

The major concern that I have on this study is the ambiguity of the definition of the shift of floral choices. When I was reading the introduction, I thought that the authors will present a study on host-plant shifts while *Eucera* species are keeping a restricted diet (i.e. shift from oligolecty to another oligolecty). For example, a study on shifts from Asteraceae pollen diet to Fabaceae pollen diet. However the shifts of diet in *Eucera* clade are also related sometimes to an increase of diet breadth (i.e. shift from oligolecty to polylecty). They are becoming more generalist. In this case, there are actually some studies already showing that extension of pollen diet increased diversification rate in Bees (Murrey et al. 2018). In an evolutionary and physiological point of view, these two kinds of shifts (specialism to another specialism, and specialism to generalism) are very different. So, authors should introduced better this concept and the study of Murrey et al. 2018. Moreover they should additionally analyze the impact of generalism in the diversification of the genus *Eucera*.

We thank the reviewer for this suggestion, which we agree is interesting. Following this suggestion, we have explored the effect of lecty (pollen diet breadth) on the diversification of eucerine bees by categorizing the species in our dataset to polylectic vs oligolectic (or pollen generalist vs pollen specialist, respectively). Our ML analyses found no significant difference between the alternative and null models, suggesting no effect of diet breadth on diversification rate of eucerine bees. Also the MCMC Bayesian analyses did not strongly support this result: while the analysis with clade-specific sampling fractions recovered the pattern of higher diversification rates of the polylectic' state, the analysis with the global sampling fractions found no effect of lecty on diversification rate.

Following the suggestion of the reviewer, we added definition to pollen specialism versus generalism in the introduction section. In line 105 we write: "Regardless of the pollen type used, pollen specialists, oligolectic species are distinguished from pollen generalist, polylectic species, which obtain pollen from a single, or more

than one botanical family, respectively (Müller and Kuhlmann 2008).” We also added a discussion of our results in lines 366-379.

In both the introduction and discussion, we refer to the study of Murray et al. 2018. In the introduction, line 88, we write: “It has been demonstrated that the transition from carnivory to pollinivory itself had not triggered diversification in bees, and subsequent broadening of host-plant diet was suspected as an important driver (Litman et al. 2011, Murray et al. 2018).” In the discussion, line 368, we write: “Such an effect corresponds with the idea that broadening of pollen host diet was a main factor that triggered diversification in the higher bees relative to primitive bee lineages, which are mostly pollen specialists (oligolectic) (Murray et al. 2018)”. We should note that the results of Murray et al. 2018 show increased diversification rate of the ‘higher’ bees relative to the oldest bee family Melittidae and the Apoid wasps, and based on this the authors concluded that the shift to pollen diet itself has not driven diversification in bees. However, their further suggestion that broadening of pollen diet must have been an important driver for diversification in bees, is an assumption that although being very likely, no analysis of pollen or floral association was done to confirm it. For example, some Melittid taxa, such as *Capicola* and *Hesperapis* in the new world and *Dasygaster* and *Melitta* in the old world, include pollen generalist (polylectic) lineages. And on the other hand, numerous lineages of other bee families are pollen specialists (oligolectic), for example the great majority of the Halictidae subfamily Rophitinae, and all of the Megachilidae subfamily Fideliinae. Certainly wide scope studies that include actual pollen data and analyses for exploring the effects on diversification in bees will be very interesting, but these are currently lacking.

Minor comments:

- Line 32: Murray et al. 2018 prove that generalism in bees increased speciation rate.

We agree that the results from Murray et al. 2018 provide intuitive support to this idea, but we don't think that they explicitly examined it and prove it, see the response above.

- Introduction: justify when you focus your study on pollen and not nectar.

Done. We added a sentence to make this clear. In lines 97 we now write: “While all species can potentially extract nectar from bee-flowers, females of different species collect different sources of pollen, which they store as food for the larvae.”

- Line 88: pollen from flower with restricted access can have particular chemical composition (as presented in the discussion). This concept should be introduced, partly to justify why you focus your study on pollen.

We thank the reviewer for this suggestion, however, we wish not to expand on this subject in the introduction section, especially because the main reason that we focus on pollen (and not nectar) as driver of diversification is that pollen choice is more strongly evolutionarily conserved than nectar. Thus, reflecting more closely the host plants of the bees, compared to nectar that could be obtained from non-host plants as well. Due to the strict limitation of word numbers, we prefer to focus our introduction on the effects of flower morphology on bee floral preference, which is the main theme of our paper and is already complex, and we feel that adding chemical composition, which is an even more complicated topic, will confuse the readers at this point.

- Line 91: evolutionary advantage of generalism? Evolutionary advantage to forage on flower with restricted access? Avoid competition? Have access to better reward?

We modified the sentence to better explain this point. In line 90 we write: “Further evolutionary innovations, namely flower buzzing (Cardinal et al. 2018) and utilisation of floral oils (Aguiar et al. 2019), were not found to

significantly increase bee diversification as well, although they contributed to increase species richness in the former and habitat occupancy in the latter.”

- Line 95: how many species of Eucera?

We have added this information in the 'taxon sample and dataset' of the methods section (line 129).

- In the results section, you mention that you considered data from literature for floral observations. You don't mention it in the M&M section. How do you make the difference between data for nectar and pollen collection?

In the results section we specifically refer to pollen hosts (rather than floral hosts in general) recorded from the literature, thus to studies that report the results of pollen analyses or pollen collecting observations, and we refer the readers to table S1 in the SM (in line 260). In this table, column 6 lists, except for pollen type fractions from our pollen analyses, published pollen analyses and pollen collecting observations. More general floral observation data are presented separately on the right column.

- Line 349: see also study on Colletes (Müller et Kuhlmann 2008)

We thank the reviewer for pointing us to this study. Because this study does not include a phylogenetic analysis, it makes it difficult to determine if shifts in floral hosts of contrasting morphologies did occur among closely related lineages.

- Line 360: Fabaceae

This previously referred to the Fabaceae tribe Fabaeae. In the revised manuscript, we refer to it with the corrected definition of IRLC (inverted repeats lacking clade), also referred to as the temperate herbaceous clade of Fabaceae.

- Lines 346– 417: I agree with this interesting section on secondary metabolites. However authors should also discuss about protein and lipid contain of Asteraceae versus Fabaceae. See the study of Vaudo et al. 2020.

- Line 384: there are a lot of study on the development of bumblebees on different diet (e.g. Vanderplanck et al. 2017; Tasei & Aupinel 2008; Moerman et al. 2016)

- Line 380: I would cite and discuss the study of Vanderplanck et al. 2017 (and not 2018).

Done. We have added a discussion on the potential effect of pollen nutrient content, including citation of relevant references as suggested (Lines 381-393).

- Figure 1: I was expecting to see the information R versus A on the figure. It would be usefull to add this information, while the code of the specimen is not really important to mention.

This information is presented on the tree using the colour of branches, black for accessible pollen, red for restricted pollen, and grey for undetermined pollen preference (see figure caption at the bottom of the main document).

- SI: host of Synhalonia acerba, restricted or accessible? I would say that access to Ericaceae pollen is restricted.

Fixed.

Additional references (not cited by the authors)

Vaudo A., Tooker J., Patch H., Biddinger D., Coccia M., Crone M., Fiely M., Francis J., Hines H., Hodges M., Jackson S., Michez D., Mu J., Russo L., Safari M., Treanore E., Vanderplanck M., Yip E., Leonard A., Grozinger C. 2020. Pollen protein: Lipid macronutrient ratios may guide broad patterns of bee species floral preferences. *Insects*, 11(2): 132.

Vanderplanck M., Vereecken N.J., Grumiau L., Esposito F., Lognay G., Wattiez R., Michez D. 2017. The importance of pollen chemistry in evolutionary host shifts of bees. *Scientific Reports*, 7 : 43058.

Moerman M., Roger N., De Jonghe R., Michez D., Vanderplanck M. 2016. Interspecific variation in bumblebee performance on pollen diet: new insights for mitigation strategies. *Plos One*, 11(12): e0168462

We thank the reviewer for referring us to these studies.

Appendix B

Associate Editor

Comments to Author:

I commend the authors for their hard work to answer all the concerns raised by both reviewers and me. I think this is much improved version, in particular with respect to explaining some of the procedures.

That said, I still think there are two aspects that need attention.

The first relates to the under-sampling and how that level of under-sampling could generate biases. The authors have now explained better their procedure and indeed some of my previous suggestions were already implemented. That said, given the level of under-sampling, and in particular our limited knowledge on how those models might misbehave in a myriad of situations that we have not fully explored, I think that a little bit of extra work needs to be done. I say this because the main result of the paper (difference in rates associated to different types of pollen) strongly rely on a statistical test where only about 20% of the species were used. Sure, the authors have taken some precautions/recommend procedures but given the level of under-sampling I think it would be necessary to run a few simulations to specifically investigate potential biases. Please see below some thoughts on that respect.

The second aspect was now raised by one of the reviewers and relates to the classification of pollen. According to the reviewer, who is a plant expert, the authors need to better describe how they came up with the restricted/accessible, oligolectic/polylectic classifications". Reviewer 1 presents a series of queries on the classification of individual specie which he felt were not classified correctly. Given that this is crucial to the main analysis, I think a better explanation of how this was done is warranted. Maybe a comparison/validation with primary literature (there must be some host plant use lists) would be a way to access the scoring of some dubious records. In that respect, a sensitive analysis (treating those dubious records differently) might be interesting, especially given the low sampling of species in the phylogeny. Few mis assignments might even change the result outcome.

We thank the editor for the constructive comments and suggestions that helped us to further improve our study. We now better address the two emphasized aspects. Specifically: (1) We reexamined all pollen samples from families that include both restricted and accessible flowers, and found only two cases of pollen types that were erroneously considered to be restricted. The concerns expressed by reviewer 1 for widely unsupported classification of pollen types stem from confused interpretation of the floral information presented in table S1, specifically between the summary lists of pollen hosts and the more general observations on floral visitation (see a more detailed response to reviewer 1 on this topic). We are now confident that our classification of pollen types is correct, and cite two extensive databases with the relevant botanical information (including both descriptions and photographs or scientific illustrations of the flower morphologies) which we used to classify flowers to contrasting pollen accessibilities; (2) We clarified the criteria used for the classification of bee species to pollen specialist vs. generalist – see our detailed response to reviewer 1 below; (3) We performed sensitivity analysis and two sets of simulations to assess the potential bias from undersampling in our dataset – please see our detailed response below.

Here I present a few thoughts and comments on the under-sampling and diversification methods that I hope will be helpful:

1) Regarding the placement of the missing species, as far as I understand there are two procedures described by Fitzjohn et al 2009. The first one they refer to "skeletal tree" and the second "terminally unresolved tree". In the first case there is a backbone tree fully resolved but incomplete with respect to sampling. Here the missing species are place within this tree. In the second case the tree "contain all

species, but their relationships are not fully resolved, with some species grouped into unresolved clades". In the case of the "terminally unresolved tree", for those "unresolved clades" there is, by definition, no crown age, because all species are collapsed into the sub-clade stem age. In that sense, yes, it can consider the possibility of missing species to be placed from the stem age because such clade lacks structure. I understood (and I could be wrong) that in your case, you had a tree with structure (fully bifurcating) where several sub-clades have more than one species, and hence it is possible to estimate a crown age for that sub-lineage. If you were using "terminally unresolved tree", then yes, by definition it is the stem age and my comment of the effect of age of crown group does not make sense (but see further comment below). On the other hand if you are using a "skeletal tree" fully resolved, I am not sure exactly how the function "make.bisse.uneven" (which I presume might have been used) allows for defining either the stem age or crown age. I took a look at the function and it seems that indeed you might be able to choose what to use (stem or crown age). Hence the authors are correct to say that the missing species can be "included" using the stem age of a defined sub-clade (node). In that respect my comment on the under-sampling affecting crown age estimates, and hence rates estimate, was indeed not fully correct. Given my misunderstanding regarding some of those issues, I suggest the authors to provide the scripts used for the diversification analysis. Apart from being very useful for reviewers and editors to understand what was exactly done, it might be useful for others in the future when trying to set up similar analysis.

We thank the editor for this comment. We now provide all scripts that were used to perform the analyses. This should allow easy reproducibility of the results and should allow readers to fully understand what was performed.

2) Although the authors are right with respect to the possibility of considering the stem age when allowing for under-sampling to be taken care, I still think that under sampling might affect rate estimates in a similar fashion because under-sampling should also affect the estimate of stem age, in particular if a missing species "intrudes" between two clades and if it that species does not belong to (was considered in the sampling fraction of) that lineages. So, although I might have been indeed incorrect to assume that diversitree forces the placement of missing species to be placed after the crown age, the same reasoning might still apply. That should be less problematic (although in practice we do not know to what extent) if the tree has all major sub-groups (e.g. all genera), but we would have to heavily rely on taxonomy to think that such problem is not relevant if we have all the sub-groups represented. I am not a bee expert so I cannot judge how the subgroups were represented, or even if we can reliably have a sense in this particular case what would be those sub-groups, how to assign the missing species to each one (I guess based on taxonomy, but....), but an under-sampling where about 80 % are missing makes me suspicious we will not run in this problem. That said you mentioned that representatives of all subgenera of *Eucera* are present. It is nonetheless an assumption, and this could be discussed as a caveat related to the under-sampling. Moreover, I think the authors should run some basic simulations to at least suggest that this level of under-sampling should not bias their results. See some suggestions below.

One of us (AD) is an expert in the taxonomy of this specific eucerine clade, and considers the possibility that some unsampled lineage would fall outside the groups already represented in our phylogeny unlikely. We added a sentence in the taxon sample and dataset of the methods section to address this concern raised. In line 130 we write: "While this selection of 89 species sums up to 22.4% of the 397 *Eucera* complex species (based on Ascher and Pickering 2021), it adequately captures the taxon diversity of this clade, with only four monotypic taxa not represented, which are not expected to alter the phylogenetic tree topology (Dorchin et al. 2018)." Dorchin et al.

(2018) list these monotypic taxa in their methods' taxon sampling section (page 83), and explain that because the endemism of these taxa to remote geographical regions, DNA samples could not be obtained. Then in Table 3 of that paper, they mention in which of the Eucera subgenera these taxa are predicted to fall based on morphology. Thus, all these taxa are predicted to fall inside the lineages already represented in our (and Dorchin et al.'s 2018) phylogeny and their inclusion will likely not have a big effect on the phylogenetic tree topology.

Regarding the effect of subsampling on the diversification analyses – please see our detailed response below (comments 4-5).

3) Still related to the under-sampling, and the use of specific sampling fraction, I am not sure you need this analysis, certainly not as the main analysis. I say that because there has been some concern in the field that using specific sampling fraction might in fact bias the results and should not be done. Here is one such opinion: <https://cran.r-project.org/web/packages/hisse/vignettes/Clade-specific-sampling.pdf>. On the light of this potential problems, I would use the specific sampling strategy as a secondary analysis, and mostly rely on the global sampling strategy for the MCMC analysis.

We thank the editor for referring us to this uncertainty in using clade specific sampling fractions. Because this type of analysis is still controversial, we decided to exclude all analyses that use clade specific sampling fractions from our study and use only global sampling fractions in the BiSSE MCMC analyses to remain congruent with the SecSSE analyses. We note that the analyses with clade-specific sampling fractions provide stronger support for higher net-diversification rates of lineages that rely on restricted pollen use, and thus the results reported in the manuscript are conservative.

4) Still related to under-sampling, I wonder on the effect of choosing equal proportions of traits for the missing species. If that is not the case, then the results could be quite different right? An under-sampling of only about 20% makes it difficult to consider if this is a reasonable assumption. Why choosing a sampling fraction of 50/50 percent? It might be seen as a conservative approach but if the frequency of R is in fact a lot smaller, then the results might be different. If I got it right, the “empirical frequencies” of those sampled species are: 47 only A, 11 that are A + R, and 22 only R. Hence if you consider “A+R” as “R”, your empirical sampling suggested that 33 out of 80 are R, about 41%. It might be interesting to run some sensitive analysis using slightly different frequencies. One could even test how skewed has the relative frequency has to be for the signal to disappear.

We thank the editor for the insightful suggestions, which we fully followed. We note that in our analyses we did not use a sampling fraction of 50/50. We used a fraction of 0.224 as a sampling fraction for each state. Since the sampling fraction, that is used for BiSSE and SecSSE analyses, represents the fraction of the observed number of taxa under a given state among the total number of species in this state, setting the fraction of each state to 0.224 assumes that the distribution of each state in the missing taxa is the same as for the observed ones. In the revised manuscript we added a sensitivity analysis that assesses the robustness of the diversification analyses to different assumptions regarding character-state distribution of the unsampled species. For each of the datasets examined, (namely PA1 ,PA2, and PA3) we repeated the diversification analyses while varying the fraction of unsampled species that are assigned to the 'accessible' state. In total, 11 different fractions were examined (from 0.0 to 1.0) and this was performed for each of the three inference methods applied (SecSSE, constrained-BiSSE MCMC, and a full BiSSE MCMC). Our results indicate that for two of the data partitions (PA2 and PA3), the results are highly robust to different assumptions regarding the state assignments of the unsampled species. Indeed, lineages in the restricted state exhibited significantly higher net-diversification rates compared to lineages

at the accessible state, unless the state distribution among the unsampled species highly deviates from that of the sampled taxa. In the PA1 data partition, the robustness of the results was more limited when using the full-BiSSE MCMC analysis. This result is congruent with the results from our power simulations (indicated below) and the empirical analyses, in which the difference in net-diversification rates between the states was statistically non-significant when using the full-BiSSE model. However, when using the constrained-BiSSE MCMC analysis, the higher diversification rate inferred for the restricted state was statistically significant. These analyses are summarized in the Methods section, lines 228–246, and a full description is provided in the supplementary materials.

5) I think the authors need to run some specific simulations to show that their level of under-sampling/frequency of traits in missing species are not biasing the results. Additional to that some model adequacy simulations might be an interesting idea. Some ideas that came to mind (really not exhaustive and you need to think this through) are:

a) Simulate a tree of similar size to what would be expected if you have all species, using the estimated rates associated to each trait, remove species to represent the level of under-sampling, assume equal trait frequencies for the missing species, and then run BiSSE analysis to see if it can indeed recover those estimates. This is a “baseline” simulation that would prove the methods works well in this space of parameters. Not the most relevant to check potential biases but nonetheless a good start.

b) Run a simulation where there is no difference in diversification associated to traits, remove species to represent the level of under-sampling, assume equal rates of traits to the missing species, assume equal trait frequencies for the missing species, and then then run BiSSE analysis to see if it can indeed recover those estimates. This would test for some kind of “false-positive”.

c) I am not sure these are the best simulations, just wanted to exemplify some possibilities and encourage the authors to think about it. The following paper might give some ideas, and be a good start: Pennell, M. W., R. G. FitzJohn, W. K. Cornwell, and L. J. Harmon. 2015. Model Adequacy and the Macroevolution of Angiosperm Functional Traits. *The American Naturalist* 186:E33–E50.

We thank the editor for these instructive suggestions, which we fully followed. We thus performed two types of simulations that are based on the specific characteristics of our data – one to assess the false positive rate and one to assess the accuracy of diversification rate estimates and the statistical power. Our results indicated that:

- A. The statistical power to correctly infer differences in diversification rates for the SecSSE and constrained-BiSSE MCMC models was high, reaching 61% and 62%, respectively. The power was lower when using the full-BiSSE MCMC model (26%), and the inferred values of the net diversification rate of state 1 (corresponding to the ‘restricted’ state) were underestimated.
- B. The results of the false-positive simulations indicated adequate false positive rate around the expected 0.05 for all the methods used, given a phylogeny and subsampling ratios similar to the empirical data.

We refer to the two sets of simulations and their results in the Methods section, in lines 228–238. A detailed description of the methods used is provided in the supplementary methods, and the results are presented in figures S2 and S3, and in table S7.

We think that the simulations performed provide further evidence that our sample fraction of 22.4% is appropriate in reliably inferring state dependent diversification rates in the eucerine clade. We emphasize these results by adding a sentence to the discussion section, in line 361: “Our results indicate a clear trend of increased diversification rate in the eucerine clade following the inclusion of restricted pollen to their diet. This trend was recovered by all analyses, even though the power to detect it was lower when using the full-BiSSE MCMC inference scheme.”

6) The revised version now has only one figure, the phylogeny with the trait information. I agree this is a very important figure. I always like to have the data in some form, so the reader is able to gain not only information, but some intuition on what the results should be, in this case seen how the branching events are distributed through time and different clades. But after the revision the second figure showing the main result (difference in diversification rate) was taken away. Given that the main result is about diversification rate differences related to trait, I think a figure showing those differences (as it was the case in the previous submission) is necessary. I would therefore suggest the authors to present a figure where we can see the diversification differences.

We have put back into the manuscript figure 2 depicting the distribution of the ratio of net diversification rates in each of the three pollen accessibility datasets and the single pollen specificity dataset used. In addition, we present the difference between the net diversification rates in each of the datasets in figure S1 as we have done in the previous version of the manuscript.

Considering those aspects, I cannot recommend the publication of this paper in its current form. If the authors think they might be able to address these issues (including simulations to show how robust the results are to this level of under-sampling, as well as other methodological choices – e.g. equal trait frequencies for missing species), then a new submission might be considered. Otherwise, I suggest adding some caveats for the diversification analysis, perhaps change the focus of the paper away from the diversification analysis, and send it to a more specialized journal.

We thank the editor for the helpful suggestions. We believe we have adequately addressed all comments and that the presented analyses are now robust.

Reviewer(s)' Comments to Author:

Referee: 2

Comments to the Author(s).

Authors did a really good job in answering to all comments / recommendations from the referees and editor. I have one last recommendation but this is not critical (that's why I proposed a minor revision). As authors considered now the level of diet specialisation in their analyses of diversification rate (great!), I think that they should do the same for the analyses of Ancestral state reconstruction. They actually already discussed about potential results in the Discussion section (line 373).

We thank the reviewer for the positive feedback. Following the reviewer's suggestion, we performed ancestral state construction analyses with the pollen specificity dataset, which we mention now in the methods section, line 181. The results from these analyses are presented in figure 1 using pie charts next to the pie charts of the pollen accessibility dataset, they are summarized in the results section, lines 314–317, and Bayes factors values are added to table S3 in the supplementary material.

Minor comments:

- Line 95: You should add a sentence to justify why you selected Eucera to explore this topic.

Done. We added the following sentence to the introduction, in line 96: "Species in this clade exhibit large variation in their floral associations, which makes them an interesting group of model."

- Line 157: I did not understand how you recorded Fabaceae. You mean, all Fabaceae have restricted pollen type? Or you did not record any non-restricted Fabaceae pollen (like from Mimosa).

We first thought that all Fabaceae species in our dataset were restricted, and following the comments we reexamined all pollen samples from families that include both restricted and accessible flowers, and reassigned them to either accessible or restricted. We have modified this sentence with a more general description, and we now write (line 158): "In plant families that include both accessible and restricted flower morphologies, we further partitioned the corresponding pollen types to either accessible or restricted." Please see additional information provided in our response on this topic to reviewer 1.

- Line 211: typo on performed

Fixed.

- Lines 269 and 275: I would say "36 species from early diverging lineages" and "44 species from recently diverging subgenera".

Agreed. We now write in line 280: "Among the 36 early diverging species sampled..."; and in line 286: "Among the 44 species sampled from the more recently diverging subgenera *Eucera* and *Synhalonia*..."

- Line 343: I did not understand the last part of the sentence.

We modified this sentence to make it clearer, and write (line 354): "This is demonstrated by our results, in bees of the genus *Melitta* (Melittidae; Dellicour et al. 2014) as well as by three studies with megachilid bees that include extensive pollen host datasets..."

Referee: 1

Comments to the Author(s).

I previously reviewed this manuscript (reviewer 1), and feel that the author's responses to my queries were largely satisfactory. I certainly defer to the editor's judgement for diversification analyses. I think the authors' inclusion of a few extra groups in the phylogeny (and not pollen data set) did little to address my concerns about completeness or ACR, but given that this is the data set that needs to be worked with – and is impressive – I do not see a real problem there. It is certainly easy to imagine that that the 80% of non-sampled species could shift, or even reverse, some of the conclusions. But that's how science works and is not a failing of the paper.

We thank the reviewer for the constructive comments and suggestions.

We agree that adding more species to our phylogeny may modify our results to some extent but we do not expect that this will have a dramatic effect such as to reverse our conclusions. This is because we included for each subgenus in our phylogeny different species groups that reflect the documented diversity of floral hosts. Floral observations for the different groups (the works cited in Table S1) can be further used as a guide. For

example, in the North American subgenus *Xenoglossodes* (= *Tetraloniella*) most of the species forage mainly on either Asteraceae or Malvaceae, and a minority of the species are associated with other botanical families, such as Fabaceae and Lamiaceae (see in LaBerge 2001). Species associated with such atypical floral hosts are included in our dataset (*X. albata* on Fabaceae and Lamiaceae and *X. salviae* on Lamiaceae and Asteraceae).

This revision, I delved into the plants, as that is where my expertise is most germane. The biggest issue is that the classifications are completely opaque – its not in the methods anywhere. This time I read through the supplement much more in depth – but not with a -toothed comb - and my biggest complaint is that you need to really describe how you came up with the restricted/accessible, oligolectic/polylectic classifications; they don't really align with what I would score from the data. There are a bunch where there were many known host families, AND you recovered multiple pollen, yet you classified them as oligolectic. Others where the fabaceous genera listed were not restricted.

I think you need to either assign uncertainty in those classifications (and bootstrap) or come up with some really strict guidelines that make sense. These will, absolutely, change the classification of some species, so the analyses will need to be rerun.

Here there has been a confusion between the pollen hosts and the more general floral observations. The pollen hosts are determined using pollen analyses and pollen collecting observations. In Table S1, this is presented in the third column to the right. They are however not determined using floral observations that are presented on the right-most column. As mentioned in the introduction section (line 98), pollen preference is a relatively conserved trait of female bees but both males and females (and other flying insects) may visit different flowers to collect nectar. Without careful observation on the behaviour of the females it cannot be determined that the bee collected pollen rather than only nectar. For example, the species *Xenoglossodes* (= *Tetraloniella*) *lippiae* that was first observed on Lippia (Verbenaceae) does not collect the pollen of Lippia, and is a well-known specialist of Malvaceae. Further, we made efforts to use only observations on females, but it is possible that some of the cited observations include also males that do not collect pollen at all. We modified the text of the caption to make this clearer.

We follow Müller and Kuhlmann (2008) in considering a species oligolectic if 95% or more of the pollen grains counted belong to the same plant family or genus. We have added a sentence in the main text on line 182 as well as in the title of Table S1 to make this clear. This explains why some of the oligolectic bee species listed in Table S1 had more than one main pollen type, when the rest of the pollen types consist less than 0.5% of the overall pollen content.

We thank the reviewer for the careful examination of Table S1 and for highlighting the errors in our classification. Following the comments, we have determined the pollen types that belong to families with both accessible and restricted flower morphologies and assigned them to either accessible or restricted pollen. In only three cases where pollen types appeared in significant amount (above 5% of the total number of grains across all samples) we were not able to determine the pollen types beyond the families Fabaceae and Lamiaceae, and excluded these pollen types from the analyses. These pollen types comprised only 7-21% of the total number of grains across all samples taken from each of three species.

To determine the floral accessibility of pollen types, we used as reference the online portal of the Kew, Royal Botanic Gardens, Plants of the World Online <http://www.plantsoftheworldonline.org/> and complemented it with information from the Flora Palaestina series (Zohary M, Feinbrun-Dothan N. 1966–1986). This latter reference is not available online but is written in English and includes useful anatomical information together with scientific

drawings that lack in the former. See for example response on the classification of *Anchusa* (Boraginaceae) bellow.

Our revisions resulted in four changes from oligolecty to polylecty, four changes from restricted to accessible+restricted pollen, and one change from accessible to accessible+restricted pollen. Please see more details bellow. Following these changes and other comments on the statistical methods, we repeated all the statistical analyses as suggested.

My other big issue is still lack of specific language.

2 - Title: I still disagree with the inclusion of flowers; there is no evidence that they diversify from bees in this paper. This title does not accurately represent the content of the paper.

We changed the title to: “Bee flowers drive macroevolutionary diversification in long-horned bees” we believe that this title represents our study accurately.

31: In your response to my previous comments, you say that you changed pollinator to floral visitor throughout. You did not; it is used incorrectly here. (i.e. in 31, you really mean – insects which utilize floral resources – pollination has nothing to do with the diversification, your paper is about floral resources). The preceding two are accurate, as you are focusing on the plant perspective.

Agreed, we changed pollinators to floral visitors in both lines 31 and 66.

34: Is it truly preference? Or just use/utilization? You don't have data on abundances to say they are preferentially using any species, just that they are using them (possibly exactly in line with abundance).

A main purpose in this study is to identify pollen host plants of the different bee species. We used pollen analyses because they can differentiate between pollen host plants that are evolutionarily conserved and other plants that the bees may occasionally visit for other purposes. It is less important for this study what exact species the bees visit, and indeed our results (in agreement with other studies) show that even oligolectic specialist species usually visit different species in the same botanical family. This information is more important for determining the actual floral hosts of the different bee species than performing field observations and calculating frequencies. We therefore think that floral preference is a more appropriate word than utilization in this sentence.

63: pollinators, again -> “those insects which utilize floral resources”

Done.

75: Why did they need to coevolve? I think you mean “evolved” here.

Agreed, we modified to ‘evolved’ as suggested.

77: Bilaterally symmetric zygomorphic flowers need not have restricted pollen, nor does concurrent diversification imply coevolution. Change the sentence before and this will be more accurate.

We agree that not all structurally modified flowers such as zygomorphic flowers are having restricted pollen, but they have more frequently than do actinomorphic flowers. We modified the sentence in line 68 to: “Flowers with concealed pollen and nectar to control the removal of rewards by foraging bees are **frequently** keel-shaped flowers...” to emphasize this.

97-98: “While all species can potentially extract nectar from bee-flowers” What does this mean? There are lots of generalized flowers that they can presumably get nectar from, too. “different sources of pollen” means specifically those plants that they do not nectar on? This section needs revision.

We modified this sentence to: “While all species may visit bee-flowers for nectar, the females of numerous species may obtain pollen from other types of flowers, and this pollen consists the main food resource for their larvae.” (line 98.) The purpose of this sentence is to provide information about the floral preference of eucerine bees, and a brief natural history background. The sentence, when read together with the two following sentences should be clearer now.

99-105: You are talking only about pollen collection here, right? Make that clearer before L 106.

That is correct. We modified the sentence copied above, to make the pollen the subject of the second part of the sentence.

105-106: “pollen specialists” belongs after “oligolectic species”, as it defines it (same with generalists).

Agreed. We modified accordingly.

109: What trait? Specialization on a single family? Or accessing restricted pollen? Ambiguous as written.

We modified this sentence and specifically mention: “How frequently has the association with restricted pollen evolved...” (line 110).

111-112: “laborious work of palynological expertise”. Nonsense as written. Do you mean “laborious work AND palynological expertise”.

We modified to this sentence and it now reads: “...primarily because of a lack of pollen host spectrum characterisation in this group of bees.” (line 112).

157: When you say only restricted pollen was recorded, do you mean you left out those species without restricted pollen when encountered or they never showed up in your dataset? They show up in your appendices as hosts, so you need to be clear here. (i.e. you have *Petalostemon*, *Dalea* as “restricted hosts” of *Xenoglossodes albata* – that’s not correct at all – though *Psorothamnus* is – since you observed them on *Dalea*, how did you differentiate that 55% fab pollen into restricted? Plus the asters are all open).

We first thought that all Fabaceae species in our dataset were restricted, and we are grateful for this important correction. We re-examined all Fabaceae pollen samples and those of other families that include both restricted and accessible flowers, and reassigned them to either accessible or restricted. We found only two cases of

Fabaceae pollen that are accessible that appeared in significant amount of more than 5% of the total pollen grains across all samples. In *Xenoglossodes albata* pollen of *Dalea* comprised 71.4% of the total pollen grains across all samples, and in *Synhalonia primiveris* pollen of *Chamaecrista* comprised 9.3%. The only other pollen type that was treated as restricted and now changed to accessible is that of *Rhododendron* (Ericaceae), but this pollen appeared in negligible amount (in *Synhalonia frater*). We have corrected the data in all tables and in the text. Interestingly, both pollen types of *Chamaecrista* and *Rhododendron* are in fact not freely accessible because these plants have poricidal anthers, such that the pollen can be collected only by bees that are capable of vibrating the anthers to release the pollen through the pores at the tip of the anthers. We nevertheless decided to treat these pollen-types as accessible to remain conservative in our assumptions and because the anthers themselves are exposed.

In the Determination of pollen type section of our manuscript we replaced the sentence referring to the Fabaceae with a more general description, and we now write in line 158: "In plant families that include both accessible and restricted flower morphologies, we further partitioned the corresponding pollen types to either accessible or restricted."

Regarding to the flowers recorded only via floral observations, listed on the right column of Table S1 (for example Asteraceae for *Xenoglossodes albata*), these are not considered as pollen hosts in our study as we explained above.

257/258: Why pollen preference? Not pollen host or something else. Preference has a very distinct definition in foraging and that is not what you are using here.

To our understanding the word 'preference' can be used to define historical association with pollen hosts in an evolutionary perspective in addition to describing foraging behaviour in an ecological perspective. Many studies cited in this work have used 'preference' for the same purpose as ours. Here are some examples:

Müller A. 1996 Host-plant specialization in western palearctic anthidiine bees (Hymenoptera: Apoidea: Megachilidae). *Ecol. Monogr.* **66**, 235–257. (doi:10.2307/2963476)

Sipes SD, Tepedino VJ. 2005 Pollen-host specificity and evolutionary patterns of host switching in a clade of specialist bees (Apoidea: *Diadasia*). *Biol. J. Linn. Soc.* **86**, 487–505. (doi:10.1111/j.1095-8312.2005.00544.x)

Müller A, Kuhlmann M. 2008 Pollen hosts of western palaeartic bees of the genus *Colletes* (Hymenoptera: Colletidae): the Asteraceae paradox. *Biol. J. Linn. Soc.* **95**, 719–733. (doi: <https://doi.org/10.1111/j.1095-8312.2008.01113.x>)

Michez D, Patiny S, Rasmont P, Timmermann K, Vereecken NJ. 2008 Phylogeny and host-plant evolution in Melittidae s.l. (Hymenoptera: Apoidea). *Apidologie* **39**, 146–62. (doi:10.1051/apido:2007048)

Sedivy C, Dorn S, Widmer A, Müller A. 2013 Host range evolution in a selected group of osmiine bees (Hymenoptera: Megachilidae): The Boraginaceae-Fabaceae paradox. *Biol. J. Linn. Soc.* **108**, 35–54. (doi:10.1111/j.10958312.2012.02013.x)

269: Its not 36 early-diverging lineages – its 36 species from early diverging lineages, right?

Agreed. We now write in line 280: "Among the 36 early diverging species sampled..."

272: Zygophyllaceae spelled wrong

Corrected.

275: there are not 44 early diverging subgenera.

True, we corrected to 'species', we now write in line 286: "Among the 44 species sampled from the more recently diverging subgenera *Eucera* and *Synhalonia*..."

288: "with high probability" belongs after inferred.

Corrected.

291-295: "preference to" -> "utilization of" (both the definition of preference, but also awkward phrasing).

As we mention above, we think that 'preference' can be used in an evolutionary context. 'Utilization' seems to describe less accurately the association between bees and their pollen hosts because these associations are far from being random. The bees do not simply utilize the flowers but they are fixed in their behaviour of utilizing their particular floral hosts. We modified this sentence to: "Neither preference of accessible, nor restricted pollen was recovered as ancestral for *Eucera* and *Synhalonia*..."

331: not foraging behaviors – resource use.

Agreed, we changed to: "Our results indicate several prominent shifts in floral host use,..."

332: delete comma

Done.

332-336: I don't understand the zygomorphy/actinomorphy part here. You are specifically talking about Fabaceae as the restricted/zygomorphic host, as borages are actinomorphic/restricted. I guess I think the symmetry is confounded with the fab/aster comparison (if Lamiaceae or Plantaginaceae were really big in the analyses, then sure, you could make a broader conclusion about symmetry, but they aren't).

We agree that there is no complete overlap between pollen accessibility, the main factor considered in our system, and flower shape. Still, the great majority of flowers with restricted pollen in our system are zygomorphic, and include not only Fabaceae, which accounted for 24% of the pollen content across all samples. Lamiaceae was collected by nine species and accounted for additional 3.8% of the overall pollen collected, and *Hygrophila* (Acanthaceae) was collected by one additional species. Restricted pollen of Boraginaceae appeared in four species that two of them are pollen specialists, and accounted for 2.7% of the pollen samples.

On the other hand, we found only three accessible pollen hosts with zygomorphic flowers: Commelinaceae appeared once, and *Dalea* and *Chamaecrista* from the Fabaceae, were each recorded once more, together accounting for 1.2% of the pollen collected. The remaining accessible pollen types that we recorded consisting 64.4% of the overall pollen collected were all of actinomorphic flowers. We would therefore prefer to still mention the distribution of pollen hosts in association to floral morphology and propose the following modification. We propose the following phrasing in line 342: "Our results indicate the occurrence of several prominent shifts in floral host use, where derived lineages incorporated restricted pollen from **structurally complex bee flowers**

into their diet. This is in contrast to ancestral lineages whose diet was based on accessible pollen from **mainly** actinomorphic flowers”

338: morphological similarity of the bees or plants? Awkward phrasing.

We meant to morphological similarity of the flowers. We modified this sentence to: “However, floral host shifts were explained in different groups of bees based on similarity in floral host shapes, ...”

346: is “higher Megachlidae” a normal phrase? I think evo. biologists (or at least botanists) are trying to get away from using higher/lower in favor of more specific/accurate phrasing (few say higher plants anymore).

Agreed. We changed to: “more recent Megachilid lineages”. (Line 359.)

375: They are “early-diverging” but since the species you sampled are modern, you can’t really say they are basal.

Agreed. We replaced “basal” with “first branching”, we now write in line 391: “...the first branching, species-poor genera *Simanthesdon* and *Protohalonia*, are mostly polylectic”.

385: pollen is known as a poor source of pollen? You mean nutrition?

Sure, we modified as suggested.

388: Delete “Especially”.

Done.

403: delete “unfavourable”

Done.

411: smaller than exactly what?

We write this explicitly now in line 427: “a smaller effect than that of pollen accessibility...”

Figure 1: italicize specific names.

We know that specific names are written italicized but in this case the names in the figure represent samples more than species, they include also the sample numbers to allow comparison with Dorchin et al. 2018, where they were first presented, and since some new names have been introduced.

Supplement:

I didn’t really want to go through this and I just skimmed it, but I guess as a plant person, I should note some that make no sense to me. They make no reasonable distinction between polylectic and oligolectic,

there are botanical errors, etc. I think they should be changed or defended and possibly the analyses rerun. Honestly, you need to proof the list and maybe look at specimens of some genera to determine accessible/restricted. There are some genera that are considered restricted in one species and accessible in another (i.e. *Dalea* in *X. albata*/*X. eriocarpi*). Also some genera are assigned to the wrong family (again, *Dalea* in *X. albata*).

If your definition of oligolectic is a single family, as is stated in L. 105-106, the assignments don't make sense. Your actual criteria needs to be stated (see my broad comments above).

***P. venusta*/*P. carinata* – why is the former polylectic and the latter oligolectic? Both have the same a single family from pollen but more known host families.**

We thank the reviewer for pointing to us these errors, but they were much fewer than thought. We explain above that our pollen diet categorization is based only on the results of our and others' pollen analyses and on pollen collection observation but not on more general floral observations, and that this has led to confusion. We have proof-read all the data in Table S1 as suggested, and fixed some botanical and spelling errors. We should mention that we did use descriptions, photographs and drawings to determine the pollen accessibility of floral hosts, and now added citations in the methods section (line 172) to the references we used. Regarding pollen lecty, we now also mention explicitly the criteria we used, which followed Müller and Kuhlmann (2008), in line 182.

***P. carinata* – lowercase specific epithet.**

Fixed.

***T. labrosa/nigripilosa* – how is this oligolectic? 17% of pollen was from other families, and other families listed in the literature.**

We thank the reviewer for this correction, this was one of four species that we reassigned from oligolectic to polylectic, see above.

***T. nigriceps* – 40% of pollen is not from asters. This is not oligolectic.**

***T. tenuifasciata* – again, how oligolectic?**

These species are both oligolectic on Asteraceae. In the previous version of our manuscript we partitioned the Asteraceae to four different pollen types that we identified, one of which is *Carduoideae*, for which we used the abbreviation CAR. For the family *Caryophyllaceae* we used the abbreviation CARY. We now aggregated all Asteraceae and use the abbreviation AST for all these types, and use CAR for *Caryophyllaceae*. This should make family abbreviation easier to follow. Please see changes made in Table S1.

***X. eriocarpin* – how is something with known hosts across 10 families oligolectic?**

This species is a good example for why floral observations can lead one astray from the true pollen preference of a bee species. This species is very common in the south-western USA and has been observed on different plants but our pollen analyses of 11 samples from 11 different sites comprised pure loads of Asteraceae pollen.

***Xenoglossodes albata*: This species is classified incorrectly as “restricted pollen” as most of the genera**

have their anthers out in the open, both the fabs (*Petalostemon/Dalea* – possibly referring the same species) and the asters (*Amorpha*, *Echinacea*) as “restricted hosts” of– that’s not correct at all – though *Psoralea* is – since you observed them on *Dalea*, how did you differentiate that 55% fab pollen into restricted? Also, *Dalea* is listed as both an aster and a fab! It’s a fab.

We thank the reviewer for this correction as this is one of the three species that collected accessible pollen of Fabaceae that we erroneously considered as restricted. Following our reexamination of pollen samples, we found that it carried either accessible pollen of *Dalea* or restricted pollen of *Stachys*, and thus changed the assignment of this species to accessible+restricted.

X. *lippii* – again why oligolectic?

This species had 97.4% pollen of Malvaceae and is therefore oligolectic on Malvaceae. Please see another comment about this species above.

***Synhalonia actiosa* has Zygothylaceae listed as pollen, but not as a host.**

As mentioned before, flowers recorded in floral observations are not considered here as pollen hosts. Following our reexamination of pollen-types, we decided to reassign the pollen-type first considered as *Larrea* from the Zygothylaceae to undetermined. This has not changed the classification of this species to accessible+restricted and polylectic.

***Synhalonia frater* – Hydrophyllum is a borage, not a fab (and it is open, not restricted). There are no Ericaceae listed.**

We corrected the erroneous family assignments in Table S1. We found that this species had 10% of *Arctostaphylos*-type pollen from the Ericaceae, and consider it as pollen host despite the lack of floral observations on Ericaceae.

***Synhalonia hamata* – Some of the borages listed are open, as are the plantages (sure they have tubes, but the pollen is accessible). Also, with that host list, how is it considered oligolectic?!? You define that in the introduction as a single botanical family; even the pollen taken off is from three families. Sure, those two fell below the 5% threshold each, but together the other families were 9%.**

As mentioned, we base our determination of diet breadth only on pollen analyses and these summed up to 92% pollen of Fabaceae. As other pollen types each comprised less than 5% they cannot be considered as legitimate pollen hosts. Mind that in every study some bee individuals are found to collect atypical pollen types in small quantities maybe due to unusual time of activity and limited availability of floral hosts. This is especially evident in spring active species such as this species.

***Eucera dafnii* – Neither restricted or oligolectic make sense with the entries. The only two genera listed are nonrestricted. Its not clear what the fab pollen came from, but I think you cannot say its oligolectic, even though the one sample had high fab pollen (also, not clear if restricted or not without a genus).**

This species is clearly oligolectic on Fabaceae based on our results of pollen analyses, and we now present the different pollen types of Fabaceae, which are all restricted.

E. proxima – again, why oligolectic? Anchusa is accessible, I've no experience with Hormuzakia.

We corrected the classification of this species to restricted and polylectic. Both *Anchusa* and *Hormuzakia* have restricted pollen because the petal leaves in these genera are fused, forming a deep tube that contains the anthers, and which entrance is blocked by five conspicuous scales. These flowers are clearly adapted to pollination by bees, mainly LT bees. See for example Müller A. 1995 Morphological specializations in Central European bees for the uptake of pollen from flowers with anthers hidden in narrow corolla tubes (Hymenoptera: Apoidea). *Entomol. Gen.* **20**,43–57.

E. nigrilabris – again, why oligolectic? Looks like a lot of hosts.

Same as above, all pollen types except for Brassicaceae appear in negligible amounts and cannot be considered as pollen hosts.

E. clypeata – why oligolectic/restricted? Caryophyllaceae is almost certainly accessible (I think even many of tubed *Silene* have the anthers everted)

In this case the claim is correct, a single specimen out of six that collected restricted pollen types of Fabaceae had large amount of Asteraceae (not Caryophyllaceae), which comprised 11.7% of the overall pollen. We corrected the classification of the species to accessible+restricted and polylectic.

Appendix C

Associate Editor:

Board Member: 1

Comments to Author:

I commend the authors for all their hard work to improve the paper. The paper is now basically ready for publication with only some minor revisions as suggested by one of the reviewers. All of those are only wording and simple additions in the text to further explain a with things.

We thank the editor and reviewers for their many helpful suggestions that helped us to further improve our study and manuscript.

Reviewer(s)' Comments to Author:

Referee: 1

Comments to the Author(s)

I'm happy with the authors' responses and their reclassification of the botanical information; I trust that the authors went through all entries and not just the ones that I highlighted.

We thank the referee for the positive feedback on our manuscript.

Referee: 3

Comments to the Author(s)

In this paper, the authors evaluate whether or not the preference for a specific type of flower changes diversification rates in eucerine bees. Despite the low sampling fraction, the authors run a series of tests to assess how their results are impacted by this lack of data, and show that their results are robust to many of this limitations. The study provides very important information not only to the knowledge of macroevolutionary patterns in insects, but also provides some insights on the role of mutualistic associations on the macroevolutionary dynamics of (at least one side of) the members of those interactions. Lastly, under a more methodological perspective, the supplementary analyses highlight that not all is lost with regards to the issues of trait-dependent diversification models, as long as researchers are careful enough to test the limitations of the results in each case.

I have very few minor comments that are described below, but the most important in my view would be to briefly address the expectations on how the prevalence between accessible and restricted pollen is in the major families of plants analysed in the study.

In general, I miss some further detail on the prevalence of bee-flowers/pollen accessible and pollen inaccessible flowers in the main families of plants, at least. This is important because even though switching to a new type of flower could represent an innovation, if there are only very few species with pollen inaccessible flowers the increase in resource might not be enough to spur diversification.

I commend the authors for the thorough exploration of the limitations in their results due to a rather small sampling fraction. It is always nice to see such care and honesty regarding the limitations in both the data and the methods, which in my opinion only make their paper more robust. The results are

encouraging as to show that it would take a large asymmetry in proportions of states within the missing species to affect their results, but I don't know much about the particular system as to say if those values are unlikely or not (as the authors mention in the second paragraph of the "Assessment of the robustness of the diversification analyses to different assumptions regarding character-state distribution of the unsampled species" section on page 3 of the SM). As mentioned in the previous comment, some further information on the expectations of this prevalence (such as a rough census of the prevalence of both types of pollen in the main families, for example) would solidify their results if they fall within the limits to which the methods look to behave well.

We thank the reviewer for the positive feedback on our analyses. We understand the rationale for presenting the prevalence of restricted vs. accessible pollen hosts of the main plant families. However, obtaining the fraction of plant species with restricted pollen is, to our knowledge, not available and demands a tedious compilation, which we believe is outside the scope of the present manuscript. Still, the use of restricted pollen type had opened a new niche for eucerine bees, which had been abundant and diverse in their geographical region of distribution. By far the richest host plant taxon with inaccessible floral morphology in our system belongs to the inverted repeats lacking clade (IRLC), comprising the great majority of the temperate herbaceous Fabaceae. This clade includes species-rich genera such as *Astragalus* (the largest plant genus with ca. 3,000 species), *Vicia* (ca. 160), *Lathyrus* (ca. 160), and *Trifolium* (ca. 250). Therefore, the utilization of this resource certainly represents an innovation that could spur rapid diversification. Following the referee's comment, we now better relate to the abundance of Fabaceae species with restricted-pollen type (Discussion lines 371-389) and we have added the number of species included in the largest genera in this clade (lines 376-379).

Lines 226-227: The posterior distribution in this case should be the distribution of differences, I didn't quite get the "percentage of MCMC samples"

We used two metrics to estimate the posterior probability that the 'restricted' state (state R) exhibits higher net diversification rate than the 'accessible' state (state A): (1) The difference in net diversification rates between the restricted and accessible states ($d_R - d_A$) and (2) The ratio of the net diversification rates of the restricted state over the accessible state (d_R / d_A). The posterior distributions are computed by recording these two estimates across all MCMC samples. Accordingly, the posterior probability that the net diversification rate under state R is higher compared to state A is the percentage of MCMC steps where the net diversification rate d_R is greater than d_A . Specifically, we compute the percentage of MCMC steps where $d_R - d_A > 0$ (the obtained value is identical when using the first or second statistic). We mention this explicitly in the Methods section and in the Results section. In lines 226-229 we write:

"We used these analyses to test whether the posterior probability for the net diversification rate of the R pollen lineages is higher than that for the A pollen lineages ($PP(d_R > d_A)$). The posterior probability that state R exhibits higher net diversification rate than state A is represented by the percentage of MCMC steps in which $d_R > d_A$."

The values relating to the second metric (d_R / d_A) are presented in figure 2, and those of the first metric ($d_R - d_A$) are presented in figure S1.

Lines 326-337: So, in this case the authors are seeing differences in speciation rate, right? It is important to think about this, because if flower type is affecting differently only the speciation rates, there is some discussion that could be made regarding the possible mechanisms behind this. Also, in relation to this, the y-axis labels in figure 2 are misleading because it shows as them being the ratio instead of the difference in rates.

Indeed, our results suggest that the effect of floral host accessibility was much greater on the speciation rate than on the extinction rate of eucerine bees. However, extinction rates are notoriously difficult to estimate and are particularly sensitive to sampling biases and departures from the assumed model (Rabosky 2010, doi: 10.1111/j.1558-5646.2009.00926.x.). Thus, the primary focus of our work was not to estimate separately speciation and extinction rates, but rather to estimate the net diversification rate ($r = \text{speciation rate} - \text{extinction rate}$), which is a more robust statistic to compute. Having said that, we have followed the reviewer suggestion and expanded the discussion to include a possible mechanism by which the flower type could affect the net diversification rate. In line 365 we write: "Utilizing restricted pollen has likely acted to increase bee net-diversification rate by opening an opportunity to exploit a vast food resource as explained below. While our analyses suggest that the difference in net-diversification rates is mainly driven by alterations in the speciation rate and not by extinction rates, these two quantities are hard to tease apart, certainly for the size of the data examined here (Rabosky 2010)."

Regarding the x-label of Figure 2, please see our answer to the previous remark above.

Lines 359-363: Couldn't the same driver be responsible for your results? It would be good to add whether or not Eucera-complex species include foreign material in nest construction as well.

Female eucerine bees, like most bees, construct nests in the ground and line their brood cell walls using hydrophobic compounds produced in dufour's gland. Members of the bee family Megachilidae lack the dufour's gland secretions and instead they use various different foreign materials for nest construction. We added the following words in bold to the sentence in line 359: "They however considered another behavioural innovation as the main driver of diversification, the inclusion of foreign material in nest construction, **the hallmark of the bee family Megachilidae.**"